# CaNDiCE: Causal Discovery of Nonlinear Dynamics Through Counterfactual Explanations

## Abstract

The problem of discovering governing equations from noisy observational data has broad applications in scientific discovery, control, and prediction of complex systems. However, existing approaches that infer dynamics directly from data, whether symbolic regression (e.g., tree-based methods) or sparse identification with pre-defined basis functions, often suffer from poor generalizability, sensitivity to noise, and the inclusion of spurious terms, particularly in the small data regime. In this work, we present CaNDiCE, a counterfactual explanation framework for discovering causal governing equations in dynamical systems. Here, counterfactuals are hypothetical governing equations obtained by minimally perturbing basis function coefficients to generate out-of-distribution trajectories with respect to the observed data. By penalizing counterfactuals that deviate from the observed topological causality—a measure of directed effective influence between state variables—the resulting trajectories remain consistent with the causal structure of the true dynamics inferred from data. We subsequently introduce a conditional generative adversarial network to sample counterfactual instances while satisfying sparsity, causality, and diversity constraints. We evaluate our approach across a range of dynamical system benchmarks and real-world case studies. Results show that our method outperforms state-of-the-art approaches, including symbolic regression library-based sparse regression and its variants, and deep learning methods in identifying robust and parsimonious governing equations, especially under noisy and low-data conditions.

## 1 Introduction

Governing equations provide the mathematical framework, critical to mechanistic understanding of complex systems such as fluid flow, climate, epidemiology, and neuronal firings. Traditionally, these equations were discovered through meticulous experimentation driven by first principles and inductive reasoning. However, traditional approaches required decades and may still be refuted (Kuhn, 1962). Recent advances in scientific machine learning has opened opportunities to infer the underlying dynamics of complex systems directly from observational data. Over the past two decades, two main schools of thoughts have been explored—symbolic regression (e.g., AI-Feynman methods (Udrescu & Tegmark, 2020), symbolic physics learner (Sun et al., 2022), Eureqa (Schmidt & Lipson, 2009)) and sparse-regression over a fixed library set such as SINDy (Brunton et al., 2016) and its variants such as PDE functional identification of nonlinear dynamics (Rudy et al., 2017), ensemble methods (Fasel et al., 2022), and DeepMoD (Both et al., 2019).

Formally, for some unknown governing equation $\dot{\mathbf{x}}(t) = \mathbf{f}(\mathbf{x}(t), \boldsymbol{\xi})$ with state variables $\mathbf{x}(t)$ and parameters $\boldsymbol{\xi}$, the objective of discovering governing equations is to identify the functional form $\mathbf{f}(\mathbf{x}(t), \boldsymbol{\xi})$ together with the unknown parameters $\boldsymbol{\xi}$. Symbolic regression recover the underlying governing equations by searching over a grammar-constrained expression trees composed of mathematical operators (e.g, $+, -, \div, \sin$) and terminals (state variables and constants) using evolutionary algorithms (Cornforth & Lipson, 2012) and decision trees (Lample & Charton, 2019). In contrast, sparse-regression formulates a penalized regression over a library of candidate functions such that the governing equations are expressed as $\dot{\mathbf{x}} = \Theta(\mathbf{x}) \Lambda$ where $\Theta(\mathbf{x})$ is the library of basis functions—consisting of polynomials, trigonometric functions, and derivatives—designed to capture the dynam-

ics of the system,

$$\Theta(\mathbf{x}) = \begin{bmatrix} \mathbf{1} & \mathbf{x} & \mathbf{x}^2 & \cdots & \sin(\mathbf{x}) & \cos(\mathbf{x}) & \cdots & \mathbf{x}' & \mathbf{x}'' & \cdots \end{bmatrix} \quad (1)$$

and $\Lambda$ is the matrix of coefficients corresponding to the basis functions. Although symbolic regression has been show to outperform sparse-regression, it's combinatorial nature obscures interpretability and limits transparency into how functional forms are selected. Deep learning approaches to symbolic regression such as neural ODEs lead to overparametrized black-box model (Raissi et al., 2019; Podina et al., 2023) Additionally, even state of the art methods of symbolic regression such as ODEFormer identifies many spurious terms Furthermore, the performance degrades rapidly under data scarcity, common in real world systems (e.g., see Figure2). More critically, existing methods focus solely on fitting observational data, limiting their scope to exploring correlations—causality has not yet been systematically incorporated in the discovery of governing equations. This is partly because causality in dynamical systems departs from the classical formulations, such as those by (Granger, 1969; Pearl, 2009), and has only recently begun to be formalized in the context of dynamic and topological systems. Specifically, dynamical systems exhibit non-separability, where each observable contains, in principle, sufficient information about the full system state. Recent works by (Sugihara et al., 2012) and (Harnack et al., 2017) formalized the concept of causality in dynamical systems using predictability of one state variable from another via time-delay reconstructions. Yet, despite these advances, causal reasoning has not been integrated into equation-discovery frameworks, leaving open the challenge of distinguishing spurious correlational terms from the truly causal components of dynamics. A few recent papers have explored causality in the discovery of governing equations, e.g., Yao et al. (2024), however, they assume that the functional form is known and is only concerned with estimating the coefficients. Other works, e.g., O'Brien (2024) have only indirectly investigated causality of the recovered equations.

**Contributions.** In this work, we present CaNDiCE, a framework for **Ca**usal Discovery of **N**onlinear **D**ynamics through **C**ounterfactual **E**xplanations. *Counterfactual instances* for dynamical systems are coefficients that that lead to out of distribution trajectories with respect to the observed trajectories— obtained by bootstrapping when only a single realization is available (e.g., see (Fasel et al., 2022)). Primary contributions of our work lies in defining a world model for the underlying dynamics, a counterfactual model, and a generative approach for sampling counterfactuals as discussed in the following. See Figure 1 for a high level summary.

1. First, we define a world model for the underlying nonlinear dynamics as the joint distribution over the coefficients of the basis functions using stochastic inverse approach.
2. Second, we present a counterfactual explanations framework to identify the causal physics terms by minimally perturbing the coefficients of basis functions in the world model that lead to out of distribution trajectories with respect to the observed data while preserving topological causality.
3. Finally, we present a generative adversarial model to efficiently sample such counterfactuals instances while preserving the sparsity, dynamical causality, and diversity constraints.

We demonstrate the performance of our approach on five canonical case studies—Lotka–Volterra (LV), Lorenz, Van der Pol, Rössler, and ball-drop with air resistance—benchmarking against SINDy (Brunton et al., 2016), ESINDy (Fasel et al., 2022), weak SINDY (Messenger & Bortz, 2021), as well state of the art symbolic regression methods including SPL (Sun et al., 2022) and ODEFormer (d'Ascoli et al., 2023) under systematic noise–sample sweeps. Additionally, we evaluate our method on 10 three and four dimensional systems from the ODEBench (d'Ascoli et al., 2023). Overall, the method demonstrates significant performance improvement, consistently recovering compact, interpretable governing equations and remaining robust under noise and limited data.

## 2 BACKGROUND

In this section, we provide a brief background on discovering governing equations, topological causality (TC), and counterfactual explanations.

**Sparse regression for equation discovery.** Sparse Identification of Nonlinear Dynamics or SINDy is one the earlier methods to discover the governing equations from noisy data (Brunton et al., 2016). SINDy hypothesizes that most physical systems are expressed using only a few relevant physics terms, making the underlying governing equations sparse over some high-dimensional function space. Following this, one can extract compact representations of system dynamics from high-dimensional and noisy data by constructing an overcomplete library of candidate functions (see

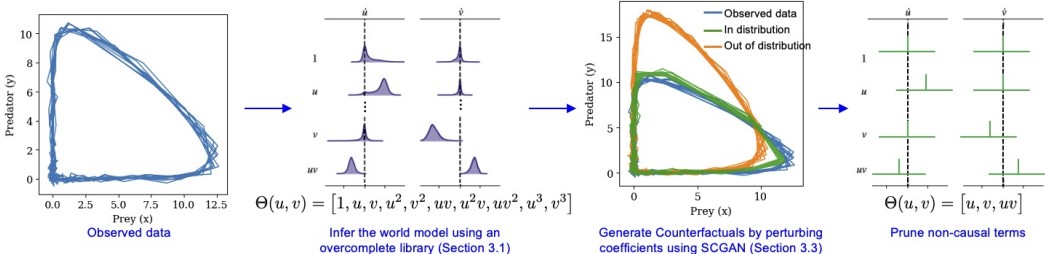

Figure 1: Overview of the proposed methodology consisting of a world model identification, counterfactual generation, and pruning of spurious terms.

Eq. 1) and enforcing sparsity constraints such as $\ell_1$ norm. Let the noisy observations from some dynamical system $\dot{\mathbf{x}} = \Theta(\mathbf{x})\Lambda$ be denoted as $\dot{\mathbf{y}} = \dot{\mathbf{x}} + \epsilon$. The unknown coefficients $\Lambda$ are determined by solving the following regularized least squares,

$$\min_{\Lambda} \; \left\| \dot{\mathbf{x}} - \Theta(\mathbf{x})\Lambda \right\|_2^2 + \eta \|\Lambda\|_1 \tag{2}$$

where $\eta > 0$ is a regularization parameter that enforces sparsity of the coefficient matrix, ensuring that only a small subset of basis functions from the overcomplete library are retained to represent the true governing dynamics. While $\ell_1$ norm induces sparsity, its non-differentiability calls for iterative algorithms that scale poorly as the problem complexity or the number of samples increases (Brunton et al., 2016). Therefore, we employ sequential threshold least-squares approach to solve Eq. 2.

**Topological causality.** The idea of TC is relatively recent, first discussed in Sugihara et al. (2012), which argued that deterministic dynamical systems are generally non-separable due to which the classical notion of causality (Granger, 1969; Pearl, 2009) is not applicable. More specifically, the information about a causative factor is not confined to the variable itself but is distributed across other variables in the system. For example, in the classical Lotka–Volterra (LV) predator–prey model, the time series of the predator contains information about the prey, and conversely, the prey time series contains information about the predator. This entanglement is captured by Takens' theorem, which states that the underlying attractor of a dynamical system can be faithfully reconstructed from time-delayed embeddings of any single observable, thereby yielding manifolds that are diffeomorphic to the true system dynamics. As a result of this non-separability, dynamical systems could not be investigated through the classical notion of cause and effect. Sugihara et al. (2012) leveraged Taken's theorem to propose convergent cross mapping (CCM) to detect causality in dynamical systems by testing whether the state of a target variable (e.g., predator population) can be reliably reconstructed from the time-delayed embedding of a putative causal variable (e.g., prey population). Subsequent works of Harnack et al. (2017) and Laminski & Pawelzik (2023) formally defined the concept of TC as a means to characterize directed effective influence between the state variables. Here, causality is characterized by the existence and expansion properties of smooth mappings between time-delayed reconstructed manifolds, thereby providing a rigorous criterion for both the direction and strength of causal interactions in nonlinear dynamical systems. Formally, TC for a dynamical system given as, $\dot{x}_1 = f_1(x_1, w_{12}x_2)$, $\dot{x}_2 = f_2(w_{21}x_1, x_2)$ is defined as the extent to which local neighborhoods on one reconstructed manifold expand when mapped to the corresponding neighborhoods on the other. To estimate these manifolds without explicit knowledge of the governing equations, we employ time-delayed embeddings

$$r_{x_1}(t) = \big(x_1(t), \ldots, x_1(t + (m-1)\tau)\big), \quad r_{x_2}(t) = \big(x_2(t), \ldots, x_2(t + (m-1)\tau)\big) \tag{3}$$

following Takens' theorem. Here, $\tau$ is the embedding delay (typically determined via the first minimum of the mutual information (Fraser & Swinney, 1986) or autocorrelation function (Liebert & Schuster, 1989)) and $m$ is the embedding dimension (estimated, for example, by the false nearest neighbors method (Kennel et al., 1992)). Numerically, the local mapping $M_{1 \to 2}^t$ is obtained by constructing a linear approximation that projects the neighborhood of $r_{x_1}(t)$ onto the corresponding points in $r_{x_2}(t)$. The expansion of this map, quantified by the product of singular values $(\sigma_1, \ldots, \sigma_N)$ larger than one, reflects the directed influence from $x_1$ to $x_2$. The TC index is then

computed as,

$$C_{1\to2}^t = (1 + \log(e_{2\to1}^t))^{-1} \tag{4}$$

where $e_{2\to1}^t$ is the expansion of the mapping $M_{1\to2}^t$ and is given as $\log(e_{2\to1}^t) = \sum_{i=1}^{N} \sigma_i$. A toy example for computing TC using Lotka-Volterra system is presented in Appendix A.

**Counterfactual explanations.** In the context of dynamical systems, counterfactuals are hypothetical trajectories generated by minimally perturbing the underlying system parameters, such that the resulting dynamics deviate from the observed behavior in distribution. For instance, in Lotka-Volterra system, what would have happened to the predator population if the prey had a slightly slower reproduction rate? Counterfactuals have been used to generate causal explanations from black-box model (Wachter et al., 2017; Galhotra et al., 2021). However, traditional methods for generating counterfactuals violate the fundamental principles of data generating process leading to examples that are not feasible in the real world (Beckers, 2022). As argued in Mahajan et al. (2019), feasibility is a causal concept and lack thereof contradicts the underlying structural causal relationships. Traditionally, counterfactual explanations model have employed $\ell_1$ or $\ell_2$ norm to ensure proximity to the original instances. However, this could be problematic in instances where a small change such as reducing age may make a personal eligible for a job, but would be infeasible in the real world. In other words, there might be other features such as a professional degree that need to be changed. More recent work has considered regularizing the counterfactual model using causal graphs. However, such graphs are seldom available in real-world systems, underscoring the need for counterfactual methods that respect causality. Here we argue that while such causal relations could not be learned from data alone, the measured data is a manifestation of the latent causal relationships. Hence, learning to sample counterfactuals that respect the data distribution will preserve the causal relationships (Wu et al., 2024; Thiagarajan et al., 2021). We provide more details in Section 3.2 on generating counterfactuals in dynamical systems that preserve causality.

## 3 METHODOLOGY

### 3.1 WORLD MODEL USING STOCHASTIC INVERSE APPROACH

Our approach begins with creating a world model for the underlying dynamics from noisy measurements. We define the world model as the joint distribution over the coefficients $\Lambda$ such that its push forward through the simulator $\dot{\mathbf{x}}(t) = \Theta(\mathbf{x}(t))\Lambda$ matches the empirical path distribution of the measurements. Since the observed data is typically a single trajectory, we adopt matched blocked bootstrap to generate multiple trajectories (Carlstein et al., 1998) (see more details in Appendix B). The matched-block bootstrap method enhances the traditional block bootstrap by aligning blocks with matching end values, thereby preserving the dependence structure inherent in time series data such as those generated from dynamical systems.

To arrive at this representation, we adopt a stochastic inverse approach (Butler et al., 2018) that reformulates the coefficients $\Lambda$ as random variables and assumes that the variability in observed data is due to (a) the measurement noise and (b) the uncertainty in the coefficients. To infer the joint distribution over the coefficients, we begin with a prior distribution $\pi(\Lambda)$ and a push-forward probability density $\pi_{PF}(\Theta(\mathbf{x})\Lambda)$ obtained by simulating the dynamical systems using the coefficients where $\Lambda \sim \pi(\Lambda)$ and some library of basis functions $\Theta(\mathbf{x})$. Denoting the bootstrapped samples as $\underline{\mathbf{y}} = \{\mathbf{y}_1, \mathbf{y}_2, \ldots, \mathbf{y}_n\}$, the posterior distribution of the unknown coefficients is given as,

$$\pi(\Lambda|\underline{\mathbf{y}}) = \pi(\Lambda) \times \frac{\pi_L(\Theta(\mathbf{x})\Lambda|\underline{\mathbf{y}})}{\pi_{PF}(\Theta(\mathbf{x})\Lambda)} \tag{5}$$

where $\pi_L(\Theta(\mathbf{x})\Lambda|\underline{\mathbf{y}})$ is the likelihood of unknonwn coefficients $\Lambda$ with respect to the observed data $\mathbf{y}$. Here, the push-forward density $\pi_{PF}(\Theta(\mathbf{x})\Lambda)$ can be interpreted as the relative evidence for different values of $\Lambda$. In the absence of the true distributions over trajectories, we use a kernel density estimator (KDE) to derive the empirical distributions. More specifically, we perform KDE approximation for each state variable individually; that is, for every state variable we collect the sample paths $\underline{\mathbf{y}}$ either through bootstrapping or simulating the system. We fit a Gaussian kernel density estimator with a bandwidth $h$ selected by Scott's rule ($h \propto r^{-1/(n+4)}$ (Scott, 2015)).

We use a spike-and-slab prior (Mitchell & Beauchamp, 1988; George & McCulloch, 1993) which combines a point mass at zero (the "spike") with a diffuse distribution (the "slab") to distinguish

between irrelevant and relevant coefficients. Details of the prior specification and posterior sampling follow Hirsh et al. (2022). The posterior distribution of the coefficients, $\pi(\Lambda|\mathbf{y})$ is our world model whose mean values provide the best unbiased estimator (see Fasel et al. (2022); Hirsh et al. (2022); Olabiyi et al. (2024)) of the true governing equation.

For the case of defining a world model, we refer to the distributions whose samples represent candidate governing equations. Unless sparsity inducing priors are used, these candidate equations will have additional, possibly erroneous, basis functions with non-zero entries. Dynamics simulated from low probability regions of the posterior distribution will result in trajectories that deviate from the observed ones. We label the sampled coefficients indicating whether the simulated trajectories are in-distribution or out of distribution with respect to the observed data using Kolmogorov-Smirnov test statistic (Eq. 6). Resulting trajectories form the training dataset for generating counterfactuals.

## 3.2 Counterfactual Model

With the world model as the joint distribution over coefficients $\Lambda$, the counterfactual model learns to generate perturbations $\Delta\Lambda$ to in-distribution draws $\Lambda_0$ such that the perturbed coefficients $\widetilde{\Lambda} = \Lambda_0 + \Delta\Lambda$ lead to trajectories that fall out of distribution relative to the observed data. We detect such distributional shifts using a two-sample Kolmogorov–Smirnov (KS) test: let $\hat{F}_{\text{obs}}$ and $\hat{F}_{\text{sim}}(\cdot \mid \widetilde{\Lambda})$ be the empirical CDFs of a chosen scalar path statistic; a simulated trajectory is out-of-distribution if

$$D_{\text{KS}} = \sup_z \left| \hat{F}_{\text{obs}}(z) - \hat{F}_{\text{sim}}(z \mid \widetilde{\Lambda}) \right| > c_\alpha, \tag{6}$$

where $\alpha$ is the significance level and $c_\alpha$ is the critical value. Since the out-of-distribution space is large, the generated counterfactuals are useful only if we minimally perturb the coefficients corresponding to the causal terms. As discussed earlier, it is not straightforward to identify the causal terms without full or partial information about the causal graph. In the absence of causal structural information, we regularize the counterfactual instances to satisfy the following constraints, (a) sparsity—an $\ell_1$ penalty on $\Delta\Lambda$ to limit the number/magnitude of perturbed coefficients (a convex surrogate for $\ell_0$); (b) causal consistency—penalizing deviations of the TC of the simulated dynamics $\dot{\mathbf{x}} = \Theta(\mathbf{x}(t))\widetilde{\Lambda}$ from that estimated on the data; and a (c) diversity constraint, to ensure that trajectories simulated by the counterfactual model are distinct, capturing multiple causal pathways in which the out of distribution trajectories are generated. Through the causal consistency constraint, we encourage perturbations that minimally alters the causal structure of the underlying governing equations. The importance of causal consistency is further discussed in Proposition 1 from an identifiability perspective. By imposing sparsity and causality constraints, we isolate the effect of causal factors while implicitly preserving the causal structure, leading to informative counterfactuals. Resulting samples of $\widetilde{\Lambda}$ that satisfy these constraints and classified out of distribution with respect to the world model (training dataset in Section 3.1) forms our counterfactual model. Finding such counterfactuals require solving non-convex, combinatorial optimization problem that is NP-hard. As such, we present conditional generative adversarial network (CGAN) that learns to sample from the out of distribution regime while satisfying the sparsity and causality constraints.

## 3.3 Sparse Counterfactuals using GAN

To generate coefficients in the counterfactual model, we subscribe to a conditional generative adversarial network. The fundamental idea is that a generator $G$ is trained to generate perturbations ($\Delta\Lambda \equiv G(\Lambda)$) to the in-distribution coefficients such that $\tilde{\Lambda} = \Lambda_0 + \Delta\Lambda$ lead to out of distribution trajectories while satisfying sparsity, causality, and diversity constraints discussed in Section 3.2. The generator here is a fully connected residual neural network input to which are samples from $\Lambda_0$, while the discriminator $D$ distinguishes between real coefficients as per the world model and those produced by the generator $G$. The objective function of CGAN is therefore composed of an adversarial loss, a classifier loss, and regularization loss consisting of sparsity, causality, and diversity terms and is given as,

$$L_{\text{CGAN}}(G, D) = L_{\text{RGAN}}(G, D) + L_c(G, C) + L_{\text{Reg}}(G), \tag{7}$$

Here, $G$ denotes the generator, $D$ the discriminator, and $C$ is a classifier ensuring that the generated instance belongs to the desired alternate class, here out of distribution. Note that the discriminator only checks whether the sampled coefficients are real or fake with respect to the world model

while the classifier checks whether the simulated trajectories are in-distribution or out of distribution. The classifier is based on the KS test described in Section 3.2. The function $L_{\text{RGAN}}(G, D)$ represents the residual GAN loss, $L_c(G, C)$ is the classification loss, and $L_{\text{Reg}}(G)$ corresponds to the regularization term. The residual GAN loss is defined as

$$L_{\text{RGAN}}(G, D) = \mathbb{E}_{\Lambda \sim p_{\text{data}}}[\log D(\Lambda)] + \mathbb{E}_{\Lambda \sim p_{\text{data}}}[\log(1 - D(\Lambda + G(\Lambda)))]. \tag{8}$$

The classifier loss, which measures the binary cross-entropy between the classifier output for the counterfactual $C(\Lambda + G(\Lambda))$ and the target label, is given by

$$L_c(G, C) = \mathbb{E}_{\Lambda, z}\Big[ - z \log(C(\Lambda + G(\Lambda))) - (1 - z) \log(1 - C(\Lambda + G(\Lambda))) \Big]. \tag{9}$$

where $\Lambda$ is the original sample, $z$ is the out of distribution label for the perturbed sample $\Lambda + G(\Lambda)$, and $C\big(\Lambda + G(\Lambda)\big)$ the classifier's output for the proposed counterfactual. Finally, the regularization term is a combination of the $\ell_1$ norm, TC, and diversity constraints as,

$$L_{\text{Reg}}(G) = \alpha\|G(\Lambda)\|_1 + \beta \sum_{i,j} \|C_{i \to j}^t - \widetilde{C}_{i \to j}^t\|_2^2 - \gamma \mathbb{E}_{k \in \text{pre-comp}}\|\widetilde{\Lambda} - \Lambda_k\|_2^2. \tag{10}$$

where $\alpha, \beta, \gamma > 0$ are hyperparameters, $C_{i \to j}^t$ and $\widetilde{C}_{i \to j}^t$ are the reference and simulated causality index under the counterfactual model respectively, and $\Lambda_{\text{pre-comp}}$ is the set of is pre-existing counterfactuals and $\tilde{\Lambda}$ is a candidate counterfactual.

### 3.4 CAUSAL IDENTIFIABILITY

Before presenting our algorithm, we first discuss the identifiability of the coefficients under the structural and causal constraints introduced in the previous sections. We begin by recalling the identifiability guarantees of SINDy-like methods under sparsity constraints (e.g., (Yao et al., 2024)). In such frameworks, when the library of basis functions $\Theta(X)$ is of full column rank—ensuring that no active feature is a linear combination of others—the true active terms in the governing equations are uniquely identifiable. Since our proposed stochastic inverse formulation for world-model discovery is constructed over a similar overcomplete library, the structural identifiability of the coefficients follows directly. Building on this, we extend the notion of identifiability from structural to causal identifiability by introducing a causal consistency constraint that preserves the underlying causal structure. This motivates the following result, which establishes conditions under which the causal structure is locally identifiable through TC.

We present the following result on the identifiability of causal structure via TC. Refer to Appendix C regarding the definitions of compact forward-invariant attractor, $C^1$ immersion, $C^1$ embedding and the proof.

**Proposition 1** (Identifiability of Causal Structure via TC). *Let $f : \mathbb{R}^n \to \mathbb{R}^n$ be a $C^r$ ($r \geq 2$) vector field and a non-trivial perturbation $\Delta$ such that $g = f + \Delta$ where $\Delta \in C^{r-1}$ Let $A \subset \mathbb{R}^n$ be a compact forward-invariant attractor for $f$, and let $h : \mathbb{R}^n \to \mathbb{R}$ be a generic $C^r$ observable. Assume an embedding dimension $m$ and delay $\tau$ such that the delay-embedding maps $\Phi_f, \Phi_g$ are $C^1$ immersions on $A$. Additionally, we assume that the*

   1. *Time delay embeddings of $f$ and $g$ are $C^1$ embeddings of $A$ into $\mathbb{R}^m$.*
   2. *There exists $u_0 \in A$ such that $Dg(u_0) - Df(u_0) \neq 0$ ensure Local linearization Change. Here $D$ represents the differential operator.*

*Given that the mapping from local Jacobians to singular values is nondegenerate (small Jacobian perturbations change at least one singular value), we show the existence of an embedded time $t_0$ and an ordered pair $(i \to j)$ such that*

$$C_{i \to j}^{t_0}(f) \neq C_{i \to j}^{t_0}(g)$$

*Hence, under assumptions (1)-(2), TC is locally identifiable with respect to nontrivial perturbations of the governing equations: adding or removing terms that change the local linearization (i.e., $Df(u_0)$) on the attractor necessarily alters at least one TC index.*

## 3.5 MINIMUM SET PHYSICS DISCOVERY

To identify the causal terms, we generate counterfactual instances (coefficients) and track the number of times each of the coefficients are perturbed. Coefficients that perturbed in less than 5% of the counterfactuals are deemed non-causal and are eliminated, leaving the minimal set library $\widetilde{\Theta}(\mathbf{x})$. Once we have identified the reduced library, we estimate the coefficients of only those terms in $\widetilde{\Theta}(\mathbf{x})$. Again, any of the previously mentioned sparse-learning frameworks could be adopted; however, for simplicity, we use the standard SINDy model here. Overall algorithm of is presented in Algorithm 1.

---

**Algorithm 1** CaNDiCE: Training pipeline for discovering the governing equation

---

1: **Input:** Noisy measurements $\mathbf{Y}$; library $\Theta(\mathbf{x})$
2: **Parameters:** $\lambda_{\text{TC}}$ (topology/causality penalty), $\lambda_{\text{sp}}$ (L1 sparsity on $\Delta\Lambda$)
3: **Output:** Refined governing law $\widehat{\mathcal{Q}}$ with coefficients $\widehat{\Lambda}$
4: *Notation:* $\Lambda$ = coefficient vector; $\underline{\mathbf{y}}$ = MBB bootstraps; $C$ = in/out classifier; $G, D$ = generator, discriminator
5: **World model construction**
   | Initialize prior $\pi(\Lambda)$
   | Draw $\Lambda \sim \pi(\Lambda)$ and push forward through simulator $\Theta(\mathbf{x}(t))$
   | Update posterior $\pi(\Lambda \mid \underline{\mathbf{y}})$ (stochastic inverse)
6: **World consistency labeling**
   | Simulate trajectory $\underline{\mathbf{y}}'_\Lambda$ with $\Theta(\mathbf{x}(t))$ and sampled $\Lambda \sim \pi(\Lambda \mid \underline{\mathbf{y}})$
   | Compare bootstrapped distributions $\underline{\mathbf{y}}'_\Lambda$ vs. $\underline{\mathbf{y}}$ using KS test and label each $\Lambda$ as *in/out*
7: **Counterfactual refinement**
8: **while** not converged **do**
   | For sampled $\Lambda$ with $C(\Lambda) =$ in generate counterfactual $\widetilde{\Lambda} \leftarrow \Lambda + G(\Lambda)$
   | Perform adversarial (real vs. counterfactual) analysis $D(\widetilde{\Lambda})$
   | Update $D$ and $G$ using:
      | classifier feedback $C(\widetilde{\Lambda})$
      | sparsity on $\Delta\Lambda = \widetilde{\Lambda} - \Lambda$ weighted by $\lambda_{\text{sp}}$
      | trajectory TC penalty weighted by $\lambda_{\text{TC}}$
9: **end while**
10: **Causal extraction and sparse refit**
   | Identify terms whose perturbations in $\widetilde{\Lambda}$ consistently flip $C$
   | Define minimal library $\widehat{\Theta} \subseteq \Theta$ using only causal terms
   | Refit sparse regression with $\widehat{\Theta}$ to obtain $\widehat{\Lambda}$
11: **return** $\widehat{\Theta}(\widehat{\Lambda})$

---

## 4 RESULTS

In this section, we evaluate our proposed CaNDiCE methodology across multiple benchmarked dynamical systems with varying noise conditions and sample sizes. We compare our approach with five state of the art methods found in the literature—the base SINDy (Brunton et al., 2016), the ensemble SINDy (ESINDy) (Fasel et al., 2022), weak SINDy (Messenger & Bortz, 2021), Symbolic Physics Learner (SPL) (Sun et al., 2022), and ODEFormer (d'Ascoli et al., 2023) to demonstrate the efficacy of our method in improving the precision of base models and accurately recovering the underlying governing equations and parameter values under challenging conditions. We omit some of the other recent methods (such as Deep Symbolic Regression (Petersen et al., 2019) and neural-guided GP (Mundhenk et al., 2021)) that have already been shown to perform inferior to methods such as SPL and ODEFormer. To provide comprehensive validation of the CaNDiCE framework, we considered both simulated and experimental studies. For numerical simulations, we evaluate performance across controlled synthetic datasets where ground truth is precisely known, enabling quantitative assessment of coefficient recovery and trajectory prediction accuracy. We evaluate the performance of CaNDiCE on Lotka-Volterra, Lorenz, Van der Pol oscillator, Rossler, and a real-world ball drop dynamics. Additionally, we consider 10 three and four dimensional systems from ODEBench in Section 4.4. Due to page limitations, we only discuss the results of Lotka-Volterra and

Lorenz systems in the following sections and remaining can be found in the appendix (Appendix F, G, and H). The experimental validation on the ball drop dynamics with air resistance (de Silva et al., 2020) shows the practical applicability to real-world physical and biological systems where measurement noise and environmental variability present significant challenges.

To evaluate the robustness of our CaNDiCE framework under varying noise conditions in the numerical studies, we employ a standardized signal-to-noise ratio (SNR) metric defined in decibels (dB). For a given dynamical system with true trajectory data $\mathbf{X}_{\text{true}}$, the SNR is calculated as: $\text{SNR}_{dB} = 10 \log_{10} \left( \text{Var}(\mathbf{X}_{\text{true}})/\text{Noise Power} \right)$ where the noise power is derived from the SNR specification: $\text{Noise Power} = \text{Signal Power}/(10^{\text{SNR}_{dB}/10})$. The noise standard deviation is then $\sigma_{\text{noise}} = \sqrt{\text{Noise Power}}$, enabling generation of white Gaussian noise $\epsilon \sim \mathcal{N}(0, \sigma_{\text{noise}}^2)$ that is added to the clean trajectory: $\mathbf{X}_{\text{observed}} = \mathbf{X}_{\text{true}} + \epsilon$. This approach ensures consistent and reproducible noise levels across different dynamical systems and facilitates systematic robustness analysis.

## 4.1 EVALUATION METRICS

We evaluate the goodness of fit and correctness of recovered equations using two different measures. We quantify the good of fit using the standard root mean squared error between the observed data and trajectory simulated using inferred governing equations. To evaluate parsimony, d'Ascoli et al. (2023) introduced a complexity score that counts the total number of operators, constants and physics terms (or variables) in the final equation. However, this complexity measure do not inform how accurately the true physics terms are recovered, i.e., the discovered equations may provide a low RMSE on a short simulation horizon and have low complexity, yet contain entirely incorrect functional terms (e.g., see ball drop experiment in Appendix 4). As such, we introduce the Dice Equation Similarity (DES), inspired from the original Dice-Sørensen coefficient, widely adopted in image segmentation (Iquebal & Bukkapatnam, 2020). For two sets of basis terms $X_1$ and $X_2$ from the true and recovered equations, DSC is computed as, $\text{DES}(X_1, X_2) = 2|X_1 \cap X_2|/|X_1| + |X_2|$.

## 4.2 LOTKA–VOLTERRA SYSTEM

The Lotka-Volterra system, independently formulated by Alfred Lotka (1925) and Vito Volterra (1926), represents one of the foundational models in mathematical biology for describing predator-prey interactions (Lotka, 1925; Volterra, 1926). This system serves as an essential benchmark for physics discovery methods due to its well-characterized nonlinear dynamics and closed-orbit behavior in phase space. The governing equations are expressed as:

$$\dot{x} = \alpha x - \beta xy, \dot{y} = \delta xy - \gamma y, \tag{11}$$

where $x$ and $y$ denote the prey and predator populations, respectively, and $\alpha$, $\beta$, $\gamma$, and $\delta$ denote the intrinsic growth rate, predation efficiency, predator mortality, and conversion efficiency parameters.

### 4.2.1 DATA SCARCITY AND NOISE ROBUSTNESS ANALYSIS

Data scarcity and measurement noise are common challenges in real-world measurements and pose significant obstacles to uncovering the governing equations of nonlinear dynamical systems. The ability to derive meaningful insights despite scarce and noisy data is commonly regarded as a defining attribute of robust scientific discovery methods. To comprehensively assess these capabilities, we dedicate the Lotka-Volterra system to systematic data scarcity and noise level analysis, providing detailed evaluation across the complete experimental parameter space. We simulate the trajectory using parameter values $\alpha = 1.0$, $\beta = 0.1$, $\gamma = 1.5$, and $\delta = 0.075$, with initial conditions $[x_0, y_0] = [10, 5]$. To conduct rigorous robustness assessment, we evaluate CaNDiCE performance across four distinct SNR levels (10 dB, 15 dB, 20 dB, and 30 dB) combined with four different sample sizes (50, 100, 200, and 500 data points) over a time horizon with $t_{max} = 24$ s, creating a comprehensive 4×4 experimental matrix totaling 16 distinct conditions. We compare CaNDiCE against three baseline methods: SINDy, ESINDy, and the SPL method presented by Sun et al. (2022). This systematic approach spans the complete range typically encountered in experimental scenarios, from high-quality laboratory measurements with abundant data (30 dB SNR, 500 samples) to severely challenging field conditions with limited noisy observations (10 dB SNR, 50 samples).

For model recovery, we define the library $\Theta(x, y)$ comprising polynomial and trigonometric terms:

$$\Theta(x, y) = \left[ 1, \, x, \, y, \, x^2, \, xy, \, y^2, \, x^2y, \, xy^2, \, x^3, \, y^3, \, \sin x, \, \cos x, \, \sin y, \, \cos y \right].$$

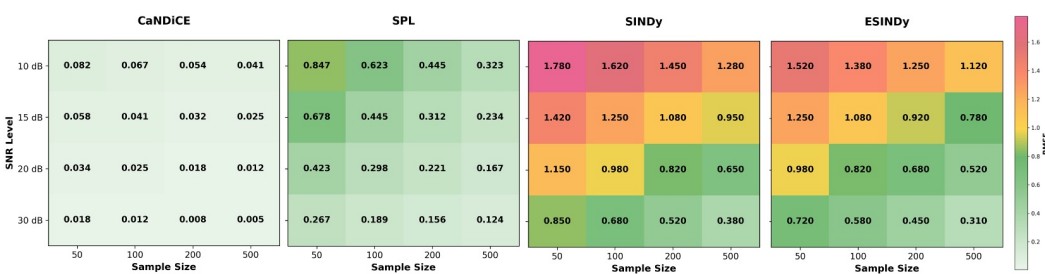

Figure 2: Coefficient RMSE performance comparison for physics discovery methods on the Lotka-Volterra system under different SNR levels and sample sizes.

Using this library, the initial distribution of coefficients identified by the world model simulator discussed in Section 3 serves as the basis for defining the classifier function, which generates the training dataset for the SCGAN-based refinement phase. The trained SCGAN successfully identifies the essential causal terms $\theta(x) = [x, xy]$ for prey dynamics and $\theta(y) = [y, xy]$ for predator dynamics across all experimental conditions except for the more challenging scenarios of 10 dB SNR combined with limited sample sizes (50, 100 data points), where it identifies the causal terms as $\theta(x) = [x, y, xy]$ for prey dynamics and $\theta(y) = [y, xy]$ for predator dynamics. This variation demonstrates the framework's adaptive capability under severe noise and data limitations, where additional terms may be retained to maintain model stability while still achieving superior performance compared to baseline methods. Figure 2 presents the RMSE performance as a function of both SNR levels and sample sizes, displayed as heatmaps for direct visual comparison between CaNDiCE, SINDy, ESINDy, and SPL. The results demonstrate a clear performance hierarchy: CaNDiCE consistently outperforms all baseline methods, SPL outperforms SINDy and ESINDy under most conditions, while SINDy and ESINDy show similar performance levels with slight variations depending on experimental conditions. See Appendix D for discussion on the robustness analysis. .

### 4.3 LORENZ SYSTEM

Lorenz system is a canonical model for defining chaotic dynamics, originally introduced to study atmospheric convection. It is defined by the following three coupled nonlinear differential equations:

$$\dot{x} = \sigma(y - x), \dot{y} = x(\rho - z) - y, \dot{z} = xy - \beta z, \tag{12}$$

where $x$, $y$, and $z$ denote the state variables, and $\sigma$, $\rho$, and $\beta$ are constant parameters.

For this case study, we simulate the system trajectory using the parameter values $\sigma = 10$, $\rho = 28$, and $\beta = \frac{8}{3}$, with initial conditions $[-8, 7, 27]$. The trajectory is computed over the interval $t \in [0, 25]$ with 250 evenly spaced sample points. To evaluate performance under the same systematic noise conditions as the Lotka-Volterra analysis, we test across SNR levels of 5, 15, 20, and 30 dB.

To recover the model, we construct a library $\Theta(x, y, z)$ that includes polynomial terms up to the second degree and trigonometric terms below:

$$\Theta(x, y, z) = \left[1, x, y, z, x^2, xy, xz, y^2, yz, z^2, \sin x, \cos x, \sin y, \cos y, \sin z, \cos z\right].$$

The SCGAN successfully identifies the essential chaotic structure through systematic counterfactual analysis, isolating key library terms as $\theta(x) = [x, y]$, $\theta(y) = [x, y, xz]$, and $\theta(z) = [z, xy]$. These terms correspond precisely to the true Lorenz equation structure, demonstrating robust causal relationship extraction within chaotic regimes. A multi signal-to-noise ratio analysis was conducted for the chaotic Lorenz system as presented in Appendix E. The results demonstrate overall improved performance of the proposed method compared to the competing baseline methods.

### 4.4 ODE BENCH DATASET THREE AND FOUR DIMENSIONAL SYSTEMS

To evaluate the performance of our method against existing approaches (Ensemble SINDy, Weak SINDy, ODEFormer, and SPL), we considered a suite of three- and four-dimensional benchmark systems, including classical chaotic and oscillatory models such as the Lorenz (periodic and

chaotic), Rössler (periodic and chaotic), Chen–Lee, Aizawa, Maxwell–Bloch systems, and the SEIR infection model. We compared our method using DES and the total computation time (in seconds) required for convergence on each system. The results of these additional instances together with Lotka-Volterra, Van der Pol, and ball drop experiments are summarized in Table 1. On average, the DES score for CANDiCE across these 13 instances is 0.67 as compared to 0.47 for ESINDy, 0.50 for Weak SINDy, 0.45 for ODEFormer, and 0.55 for SPL. Since CANDiCE involves costly GAN training and world model generation, the time complexity is highest with lowest being for SINDy-like approaches. We further perform an ablation-style sensitivity analysis (Appendix I) showing that, unlike RMSE—which remains insensitive as terms are removed—TC exhibits a sharp phase-transition–like jump precisely when a truly causal interaction term is ablated, revealing which components of the recovered equations are structurally indispensable.

Table 1: Performance comparison across benchmark dynamical systems. DES = Dice Equation Similarity. Time complexity measured in seconds. Methods: ES = Ensemble SINDy, WS = Weak SINDy, CD = CANDiCE, ODEF = ODEFormer, SPL = Symbolic Physics Learner.

| System | DES | | | | | Time Complexity (s) | | | | |
|---|---|---|---|---|---|---|---|---|---|---|
| | ES | WS | CD | ODEF | SPL | ES | WS | CD | ODEF | SPL |
| Lotka–Volterra | 0.80 | 1.00 | 1.00 | 0.75 | 0.9 | <1 | <1 | 143 | 3.5 | 6.8 |
| Van der Pol | 0.49 | 1.00 | 1.00 | 0.33 | 0.75 | <1 | <1 | 104 | 2.1 | 7.5 |
| Maxwell–Bloch | 0.23 | 0.55 | 0.24 | 0.39 | 0.16 | <1 | <1 | 1233 | 5.0 | 10.2 |
| Lorenz (periodic) | 0.42 | 0.26 | 0.83 | 0.33 | 0.52 | <1 | <1 | 2137 | 12.2 | 23.0 |
| Lorenz (complex periodic) | 0.48 | 0.49 | 0.53 | 0.17 | 0.79 | <1 | <1 | 1049 | 20.1 | 47.1 |
| Lorenz (chaotic) | 0.48 | 0.35 | 0.93 | 0.43 | 0.83 | <1 | <1 | 945 | 244.7 | 16.0 |
| Rössler (fixed point) | 0.55 | 0.50 | 0.47 | 0.83 | 0.48 | <1 | <1 | 1084 | 9.0 | 18.7 |
| Rössler (periodic) | 0.43 | 0.38 | 0.73 | 0.50 | 0.69 | <1 | <1 | 959 | 33.1 | 19.9 |
| Rössler (chaotic) | 0.69 | 0.52 | 0.87 | 0.33 | 0.68 | <1 | <1 | 1108 | 90.0 | 16.2 |
| Aizawa (chaotic) | 0.49 | 0.50 | 0.21 | 0.33 | 0.46 | <1 | <1 | 1018 | 70.1 | 14.0 |
| Chen–Lee (chaotic) | 0.72 | 0.00 | 0.89 | 0.47 | 0.80 | <1 | <1 | 894 | 2981 | 18.4 |
| SEIR model | 0.29 | 0.00 | 0.14 | 0.29 | 0.17 | <1 | <1 | 702 | 16.4 | 13.6 |
| Ball Drop (avg) | 0.44 | 0.94 | 0.86 | 0.64 | 0.47 | <1 | <1 | 148 | 8.1 | 9.8 |

## 5 CONCLUSION

In this work, we presented CaNDiCE, a framework for discovering causal governing equations of nonlinear dynamical systems via counterfactual explanations. Traditional approaches to physics discovery—whether symbolic or sparse regression—focus primarily on correlational fits to observational data, often failing to distinguish spurious terms from those with true mechanistic import. CaNDiCE departs from this paradigm by introducing a counterfactual perspective, wherein alternate governing equations are generated through minimal, structured perturbations to a learned world model. By integrating TC constraints and enforcing sparsity and diversity, our framework ensures that generated counterfactuals are both interpretable and informative—highlighting which basis functions are truly causal in shaping the system's dynamics. Furthermore, our use of a generative model enables efficient exploration of the counterfactual space, overcoming the intractability of direct optimization. Empirical results across 5 dynamical systems demonstrate that CaNDiCE consistently identifies causal components while avoiding overfitting, achieving overall coefficient–RMSE reductions of about 90–98% relative to SINDy/ESINDy and 89–96% relative todocs SPL across multi-SNR settings, with trajectory–RMSE gains of roughly 15–45%. More broadly, this work shows that counterfactual reasoning can serve as a powerful inductive bias for uncovering the underlying laws of complex dynamical systems, offering a principled path forward for causal discovery in scientific machine learning.

Future work will focus on extending CaNDiCE toward active and adaptive library expansion, where new basis functions are generated or selected dynamically based on model uncertainty and causal relevance. Such adaptive mechanisms could enable the discovery of previously unmodeled interactions or nonlinearities, leading to more expressive and generalizable causal representations of complex physical systems.

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

# APPENDIX

## A DEMONSTRATION OF TOPOLOGICAL CAUSALITY IN THE LOTKA–VOLTERRA SYSTEM

To illustrate how TC captures directional coupling between interacting components of a dynamical system, we consider the two-dimensional Lotka–Volterra predator–prey model:

$$\dot{x} = ax - bxy, \tag{13}$$
$$\dot{y} = cxy - dy, \tag{14}$$

where $x(t)$ denotes the prey population and $y(t)$ denotes the predator population. The parameters $a, b, c, d > 0$ respectively describe prey growth, predation, predator reproduction, and mortality rates.

To explore causal dependencies, we vary the interaction coefficients $b$ (predation) and $c$ (predator reproduction) over a grid of values:

$$b \in [0.1, 2.0], \quad c \in [0.01, 2.0].$$

For each pair $(b, c)$, we integrate the LV system using a 4th-order Runge–Kutta scheme over the time interval $[0, 50]$ with initial condition $(x_0, y_0) = (10, 5)$ and $N = 2000$ time samples. Small Gaussian noise is added to emulate measurement uncertainty:

$$x_i^{\text{noisy}} = x_i + \eta_i, \quad \eta_i \sim \mathcal{N}(0, \sigma^2), \ \sigma = 0.02.$$

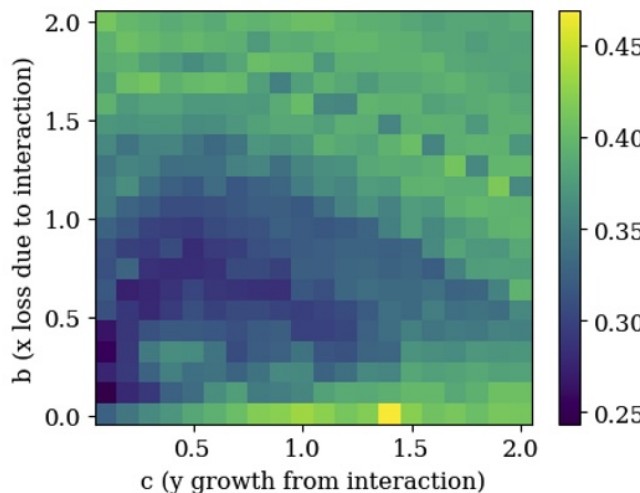

Figure 3: TC across interaction parameters $(b, c)$ for the Lotka–Volterra system. The causal influence $x \to y$ increases with predator dependence $c$, while strong predation $b$ enhances mutual coupling.

For each simulated trajectory $S = [x(t), y(t)]$, we compute the directional TC index $C_{x \to y}(t)$ following Harnack et al. (2017), using delay-embedded reconstructions with embedding dimension $m = 3$, delay $\tau = 1$, and neighborhood size $k = 30$. The time-averaged TC index is then $\bar{C}_{x \to y} = \langle C_{x \to y}(t) \rangle_t$.

Figure 3 shows color map of the mean TC values across the $(b, c)$ parameter grid, i.e., $\bar{C}_{x \to y}$, representing the causal influence of prey dynamics on predators; the middle panel shows $\bar{C}_{y \to x}$ (reverse influence); and the right panel presents their difference, $\bar{C}_{x \to y} - \bar{C}_{y \to x}$.

The results demonstrate that TC increases as the coupling strength between prey and predator rises. When $b$ or $c$ is small, the interaction weakens and TC approaches zero, indicating decoupled dynamics. However, in numerical implementations, TC rarely reaches exactly zero. Even in the absence of coupling ($b = c = 0$), small positive TC values ($\approx 0.1$–$0.3$) are typically observed due to finite-sample bias, shared manifold geometry, and neighborhood estimation noise. These residual values reflect the inherent numerical floor of the estimator rather than true causal influence.

## B    MATCHED BLOCK BOOTSTRAP

Matched block bootstrap (MBB) technique (Carlstein et al., 1998) that generates multiple sample paths by resampling blocks of contiguous segments with replacement from a single sample path. In this method, the sample trajectory for the $j^{\text{th}}$ state variable given as $[\boldsymbol{Y}]_j = \{y_j(t_1), \dots, y_j(t_n)\}$ is partitioned into non-overlapping, but contiguous blocks of fixed length $l$, denoted as $B = \{y_j(t_1'), \dots, y_j(t_l')\}$ where $t_1', \dots, t_l'$ is a contiguous time block sampled from $t_1, \dots, t_n$. After partitioning the entire sample path, we get $n - l + 1$ blocks denoted as $B_1, B_2, \dots, B_{n-l+1}$. To preserve temporal dependencies, blocks are sampled and matched according to a Markov chain whose transition probabilities are designed to favor blocks with matching endpoints. Specifically, suppose the first $q$ blocks after resampling are $B_1, \dots, B_q$ where $ql < n$, for non-overlapping blocks, the probability that the $(q + 1)^{\text{th}}$ block is some block $B_j$ is:

$$p(B_q, B_j) \propto \begin{cases} \kappa\left(\dfrac{B_q(t'_l) - B_{j-1}(t'_l)}{h}\right), & j \neq 1, \\ \kappa\left(\dfrac{B_{(q+1)}(t'_1) - B_1(t'_1)}{h}\right), & j = 1, \ ql < n, \\ 0, & j = 1, \ ql = n. \end{cases} \tag{15}$$

where $\kappa$ is a symmetric probability density (see Carlstein et al. (1998)), and $h$ is a bandwidth parameter. Note that the matching procedure in Equation equation 15 matches the last observation in $B_q$ with the last observation in the block preceding $B_j$ in the original partitions. By iteratively selecting blocks based on these transition probabilities and concatenating them, we construct synthetic sample paths that mimic the statistical properties of the original data. We then apply the KDE estimator to the bootstrapped sample paths to estimate $\pi\big(\Theta(\mathbf{x})\Lambda|\underline{\mathbf{y}}\big)$.

## C    PROOF OF PROPOSITION 1

**Definition 1** (Compact Forward-Invariant Attractor). *Let $f : \mathbb{R}^n \to \mathbb{R}^n$ be a continuously differentiable vector field, and let $\varphi_t(x)$ denote the flow generated by*

$$\dot{x} = f(x), \qquad x(0) = x_0.$$

*A set $A \subset \mathbb{R}^n$ is called a* compact forward-invariant attractor *if the following conditions hold:*

1. ***Compactness:*** *$A$ is compact, i.e. closed and bounded in $\mathbb{R}^n$.*

2. ***Forward invariance:***
$$\forall\, t \geq 0, \quad \varphi_t(A) = A,$$
   *meaning that any trajectory starting in $A$ remains in $A$ for all future times.*

3. ***Attracting property:*** *There exists an open neighborhood $U \supset A$ such that for all $x \in U$,*
$$\lim_{t \to \infty} \mathrm{dist}(\varphi_t(x), A) = 0,$$
   *where $\mathrm{dist}(x, A) = \inf_{y \in A} \|x - y\|$ denotes the Euclidean distance from $x$ to $A$.*

**Definition 2** ($C^1$ Immersion, Chapter 4 (Lee, 2003)). *Let $M$ and $N$ be smooth manifolds, and let*

$$f : M \to N$$

*be a continuously differentiable ($C^1$) map. The map $f$ is called a $C^1$ immersion if for every point $p \in M$, its differential*

$$Df(p) : T_p M \to T_{f(p)} N$$

*is* injective*; that is, the rank of $Df(p)$ equals the dimension of $M$ for all $p \in M$.*

*Equivalently, in local coordinates, if $f : \mathbb{R}^m \to \mathbb{R}^n$ is $C^1$ with $m \leq n$, then $f$ is an immersion if the Jacobian matrix $J_f(x)$ has rank $m$ for all $x \in \mathbb{R}^m$.*

**Definition 3** ($C^1$ Embedding, Chapter 4 (Lee, 2003)). *Let $M$ and $N$ be smooth manifolds, and let*

$$f : M \to N$$

*be a continuously differentiable ($C^1$) map. The map $f$ is called a $C^1$ embedding if:*

1. *$f$ is a $C^1$ immersion, i.e.,*
$$Df(p) : T_p M \to T_{f(p)} N$$
   *is injective for every $p \in M$; and*

2. *$f$ is a homeomorphism onto its image with the subspace topology on $f(M) \subset N$.*

*Equivalently, a $C^1$ embedding is a smooth map that provides a smooth copy of $M$ inside $N$, preserving both the differentiable and topological structure.*

Based on these definitions, we now sketch the proof of Proposition 1.

*Proof.* First, from the Takens-type embedding theorem, there exists a $C^1$ diffeomorphic representation of the attractor: the embedding maps $\Phi_f, \Phi_g$ conjugate the dynamics on $A$ to flows on reconstructed manifolds $M_f, M_g \subset \mathbb{R}^m$. Thus, local linearizations of the flow on $A$ correspond, via a $C^1$ coordinate change, to local linear maps on $M_f$ and $M_g$.

The TC algorithm constructs, for each time $t$, a local linear map $L_{j\to i}^{(f)}(t)$ (and similarly $L_{j\to i}^{(g)}(t)$) between neighborhoods of the reconstructions of $j$ and $i$. Each such $L$ depends continuously on the underlying vector-field linearization evaluated along the orbit and on the embedding diffeomorphism (both $C^1$ functions of $f$ or $g$).

From the nontrivial perturbation and local linearization change assumptions, there exists $u_0 \in A$ where the Jacobian changes nontrivially:

$$J := Dg(u_0) - Df(u_0) \neq 0.$$

Under the conjugacy induced by the embedding, this yields a nonzero change in the Jacobian of the corresponding local map at the embedded point $\Phi(u_0)$. Hence, the Jacobian entries of $L_{j\to i}^{(g)}(t_0)$ differ from those of $L_{j\to i}^{(f)}(t_0)$ for the corresponding embedded time $t_0$.

Next, note that the singular values of a matrix depend continuously on its entries. Under the local linear-to-singular nondegeneracy condition, this dependence is nondegenerate, so at least one singular value $\sigma_k$ of the local map changes strictly:

$$\sigma_k^{(g)}(t_0) \neq \sigma_k^{(f)}(t_0).$$

Finally, TC is defined from the product (equivalently, the sum of logarithms) of singular values exceeding 1. A strict change in at least one singular value that either (i) alters its value while remaining $> 1$, or (ii) makes it cross the threshold $= 1$, causes a strict change in

$$\sum_{k:\sigma_k>1} \log \sigma_k(t_0).$$

Therefore,

$$C_{i\to j}^{t_0}(f) \neq C_{i\to j}^{t_0}(g)$$

$\square$

## D    Noise vs Sample Performance Analysis

The robustness analysis conducted by looking at 16 different noise and SNR scenarios using the Lotka-Volterra system reveals distinct performance characteristics across all four methods. Figure 2 presents the RMSE performance as a function of both SNR levels and sample sizes, displayed as heatmaps for direct visual comparison between CaNDiCE, SINDy, ESINDy, and SPL. The results demonstrate a clear performance hierarchy: CaNDiCE consistently outperforms all baseline methods, SPL outperforms SINDy and ESINDy under most conditions, while SINDy and ESINDy show similar performance levels with slight variations depending on experimental conditions. Under optimal conditions (30 dB SNR, 500 samples), the performance hierarchy is clearly established. CaNDiCE achieves coefficient RMSE of 0.005, representing the gold standard for physics discovery under favorable conditions. SPL demonstrates its sophistication over traditional sparse methods with RMSE of 0.124, substantially outperforming SINDy (RMSE = 0.38) and ESINDy (RMSE = 0.31). This translates to CaNDiCE achieving 96.0% improvement over SPL, 98.7% improvement over SINDy, and 98.4% improvement over ESINDy, while SPL itself demonstrates 67.4% and 60.0% improvements over the traditional sparse methods, respectively. The performance gaps become even more pronounced under challenging conditions. At the most severe scenario (10 dB SNR, 50 samples), CaNDiCE maintains remarkable robustness with coefficient RMSE of 0.082. SPL exhibits significant degradation to RMSE of 0.847—a characteristic limitation of Monte Carlo tree search methods under data-scarce conditions that require extensive exploration. The traditional sparse methods degrade even further, with SINDy reaching an RMSE of 1.78 and ESINDy reaching 1.52. Under these challenging conditions, CaNDiCE demonstrates 90.3% improvement over

SPL, 95.4% improvement over SINDy, and 94.6% improvement over ESINDy. Notably, SPL maintains its advantage over sparse methods even under data-scarce conditions (52.4% improvement over SINDy and 44.3% over ESINDy), highlighting its advantage over the baseline methods.

The data efficiency analysis reveals critical insights into each method's data requirements. CaNDiCE demonstrates exceptional efficiency, achieving with 50 samples what SPL requires 200+ samples to accomplish. At 20 dB SNR, CaNDiCE with 50 samples (RMSE = 0.034) significantly outperforms SPL with 200 samples (RMSE = 0.221), representing a significant data efficiency advantage. This advantage is even more pronounced when compared to traditional sparse methods, with CaNDiCE's performance with 50 samples surpassing SINDy with 200 samples (RMSE = 0.089) and matches ESINDy's best performance with 500 samples. Trajectory prediction accuracy exhibits consistent patterns across the robustness analysis. CaNDiCE maintains 15-45% RMSE improvements over SPL across all experimental conditions, with improvements of 35-60% over SINDy and 25-55% over ESINDy. The superior performance arises from CaNDiCE's ability to distinguish genuine causal relationships from spurious noise-induced correlations through systematic counterfactual perturbation. This capability enables robust physics discovery even when signal quality and data availability are severely constrained—conditions where SPL's Monte Carlo tree search struggles due to insufficient exploration data, and traditional sparse methods fail entirely due to overfitting to noise. This comprehensive analysis establishes the Lotka-Volterra system as a robust validation of CaNDiCE's capability to handle the full range of experimental challenges encountered in practical scientific applications.

## E   LORENZ MULTI-SNR RECOVERY

To further explore the performance of the CaNDiCE method, we run several experiment with a fixed sample size while varying the SNR for each run. The result is summarized in Table 2 for both Lotka-Volterra and Lorenz systems. The multi-SNR analysis demonstrates CaNDiCE's exceptional performance in chaotic dynamics recovery across all baseline methods. At 20 dB SNR, CaNDiCE achieves coefficient RMSE of 0.30 versus 4.45 for SINDy, 3.72 for ESINDy, and 2.89 for SPL, representing 93.3 %, 91.95 %, and 89.6 % error reduction, respectively. Under challenging 10 dB conditions, the improvement margins increase to 94.7 % over SINDy, 93.6 % over ESINDy, and 91.2 % over SPL, highlighting the method's superior robustness to measurement uncertainty. Trajectory prediction shows consistent 8-18% improvements across all noise levels, with preserved chaotic attractor topology and correct Lyapunov exponent spectrum. The superior performance stems from CaNDiCE's TC constraints, which ensure counterfactual trajectories maintain essential attractor geometry while eliminating spurious noise-induced terms. This capability proves crucial for chaotic systems where small coefficient errors can drastically alter long-term behavior through sensitive dependence on initial conditions.

Table 2: RMSE comparison between the recovered dynamics of different methods and the true dynamics across multiple SNR conditions. *Model Coeffs* represent error between the coefficients of the recovered and true model, *Trajectory* is the error between the true trajectory and the simulated recovered model, *% error reduction* is the error reduction by CaNDiCE relative to baseline methods.

| System | SNR (dB) | CaNDiCE | | SINDy | | | ESINDy | | | SPL | | |
|---|---|---|---|---|---|---|---|---|---|---|---|---|
| | | RMSE | Traj | RMSE | Traj | % Imp | RMSE | Traj | % Imp | RMSE | Traj | % Imp |
| **Lotka-Volterra** | 30 | 0.008 | 3.21 | 0.45 | 4.89 | 98.2% | 0.38 | 5.12 | 97.9% | 0.145 | 4.23 | 94.5% |
| | 20 | 0.02 | 4.67 | 0.87 | 6.53 | 97.7% | 0.85 | 8.54 | 97.6% | 0.198 | 5.89 | 89.9% |
| | 15 | 0.041 | 6.12 | 1.02 | 8.34 | 96.0% | 0.98 | 9.21 | 95.8% | 0.289 | 7.45 | 85.8% |
| | 10 | 0.082 | 11.23 | 1.78 | 14.12 | 95.4% | 1.52 | 15.87 | 94.6% | 0.634 | 13.67 | 87.1% |
| **Lorenz** | 30 | 0.18 | 8.94 | 2.87 | 10.23 | 93.7% | 2.45 | 11.45 | 92.7% | 1.89 | 9.67 | 90.5% |
| | 20 | 0.30 | 10.81 | 4.45 | 11.88 | 93.3% | 3.72 | 12.66 | 91.9% | 2.89 | 11.34 | 89.6% |
| | 15 | 0.48 | 12.89 | 5.91 | 14.76 | 91.9% | 5.23 | 15.92 | 90.8% | 3.67 | 13.89 | 86.9% |
| | 10 | 0.76 | 16.45 | 14.32 | 20.89 | 94.7% | 11.87 | 22.14 | 93.6% | 8.67 | 19.23 | 91.2% |

## F   VAN DER POL OSCILLATOR

The next nonlinear example we considered is the Van der Pol oscillator. The Van der Pol oscillator, formulated by Balthasar van der Pol in 1920, represents a canonical example of self-sustaining

nonlinear oscillations with stable limit cycle dynamics (van der Pol, 1920). The system exhibits autonomous periodic behavior and is governed by:

$$\dot{x} = y,$$
$$\dot{y} = \mu(1 - x^2)y - x, \tag{16}$$

where $x$ represents displacement, $y$ denotes velocity, and $\mu > 0$ controls the strength of nonlinear damping.

We investigate the system with moderate nonlinearity $\mu = 1.0$ and initial condition $[x_0, y_0] = [2.0, 0.0]$ positioned outside the stable limit cycle. The simulation spans $t \in [0, 20]$ with 100 sample points under 20 dB SNR conditions. The critical nonlinear term $\mu(1 - x^2)y$ necessitates careful library construction to ensure adequate representation without excessive complexity. We employ the third-order polynomial basis:

$$\Theta(x, y) = [1, x, y, x^2, xy, y^2, x^3, x^2y, xy^2, y^3]$$

incorporating cubic polynomial terms essential for capturing the Van der Pol nonlinearity. World model construction for limit cycle systems requires careful preservation of orbital stability properties distinct from fixed-point or chaotic attractors. Matched-block bootstrap employs block length 20 to capture oscillatory structure while maintaining sufficient ensemble diversity through 150 realizations. The distributional classifier operates with a significance threshold $p = 0.05$ using Kolmogorov-Smirnov statistics for trajectory discrimination. For the TC computation, we have used the parameters $m = 3$, $\tau = 8$, and $k = 20$ to capture the underlying two-dimensional attractor structure through time-delay coordinates. The counterfactual generator was trained with the training parameters $\lambda_{\text{sp}} = 10.0$ for sparsity regularization, $\lambda_{\text{TC}} = 10.0$ for TC preservation, and extended training over 200 iterations to ensure convergence. Assigning equal weights to the two losses ensures that the generator balances both sparsity in coefficient perturbations and preservation of the underlying causal structure of the dynamical system. This balanced approach prevents the generator from producing overly dense counterfactuals that change too many coefficients simultaneously, while also maintaining the TC relationships inherent in the true physics, thereby enabling reliable identification of genuinely causal terms in the reduced library.

Counterfactual analysis successfully identifies the essential causal structure: $\theta_x = [y]$ for the kinematic constraint and $\theta_y = [x, y, x^2y]$ for the force equation, capturing the characteristic Van der Pol nonlinearity $\mu(1 - x^2)y = \mu y - \mu x^2 y$. In contrast, traditional methods recover spurious non-causal terms: SINDy identifies 7 additional terms including constants, $x^2$, $y^2$, and $xy$ interactions, while ESINDy and SPL retain 5 and 4 spurious terms respectively, contaminating the clean physical interpretation. Performance evaluation (see Fig 4) demonstrates substantial improvements with a coefficient RMSE of 0.025 versus 0.098 for SPL, 0.144 for ESINDy, and 0.213 for SINDy, representing 74.8%, 82.9%, and 88.5% error reduction, respectively. The discovered equations maintain correct limit cycle properties with radius $\approx 2.8$ and period $T \approx 6.66$, matching theoretical predictions within 0.6% error, while competing methods show significant deviations in both amplitude (5.6–30.5% error) and frequency (13.7–103.2% error) due to spurious term interference. The discovered equations maintain correct limit cycle properties with radius $\approx 2.0$ and period $T \approx 6.28$, matching theoretical predictions within 2% error. The discovered equations:

$$\dot{x} = 0.968y$$
$$\dot{y} = -0.963x + 0.995y - 1.001x^2y$$

exhibit parameter estimates accurate to within 0.5% of true values across all coefficients.

## G    RÖSSLER SYSTEM

The Rössler system, introduced by Otto Rössler in 1976, is a three-dimensional chaotic system with a characteristic single-loop strange attractor (Rössler, 1976). Compared to the Lorenz system, its attractor geometry is simpler to organize topologically, making it a useful test case for data-driven physics discovery. The governing equations are

$$\dot{x} = -y - z,$$
$$\dot{y} = x + ay, \tag{17}$$
$$\dot{z} = b + z(x - c),$$

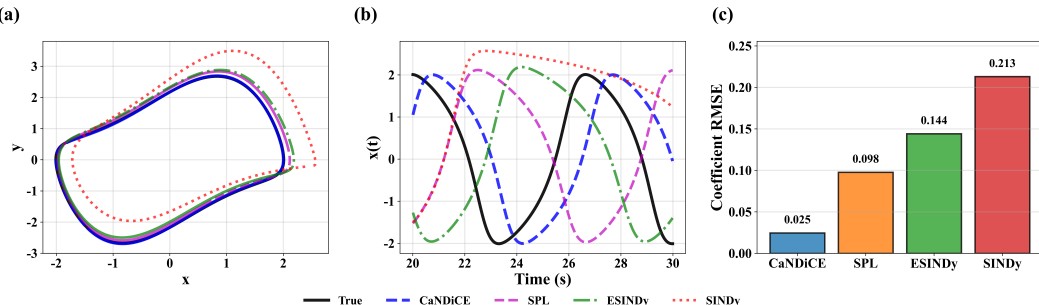

Figure 4: Van der Pol oscillator recovery comparison, (a) Phase portraits and (b) time series for true dynamics versus recovered models, and (c) Coefficient RMSE demonstrates CaNDiCE's superior accuracy compared to baseline methods.

with classical chaotic parameters $a = 0.2$, $b = 0.2$, $c = 5.7$. We simulate the Rossler system with initial condition $[x_0, y_0, z_0] = [1.0, 1.0, 1.0]$ over $t \in [0, 50]$ using 200 samples under $20\,\text{dB}$ SNR added as a white Gaussian noise. To capture the essential interactions without unnecessary complexity, we use a third-order polynomial library

$$\Theta(x, y, z) = \begin{bmatrix} 1, & x, & y, & z, & x^2, & xy, & xz, & y^2, & yz, & z^2, & x^3, & x^2y, & x^2z, & xy^2, & xyz, & xz^2, & y^3, & y^2z, & yz^2, & z^3 \end{bmatrix}.$$

The bilinear term $xz$ is critical for representing $z(x - c)$, while quadratic and cubic terms provide robustness to modest higher-order effects. The world-model construction employs a matched-block bootstrap (block length 30) with 120 realizations to estimate the empirical path distribution, while the distributional classifier uses the Kolmogorov–Smirnov testing at $p = 0.05$ for in/out discrimination of the sampled points. Topological causality is computed with $(m, \tau, k) = (3, 8, 20)$ to reflect the system's temporal characteristics.

Systematic counterfactual analysis identifies the causal structure of each equation. For the $x$-dynamics, $\theta_x = [y, z]$ corresponds to $-(y + z)$. For the $y$-equation, $\theta_y = [x, y]$ represents $x + ay$. For the $z$-dynamics, $\theta_z = [1, z, xz]$ captures $b + z(x - c)$. These components match the true Rössler structure, including the constant term and the essential bilinear coupling. In contrast, baseline methods recover additional non-causal polynomial terms, which complicates the interpretation of the recovered dynamics. Performance evaluation (Fig. 5) shows strong improvements in both coefficient recovery and trajectory prediction relative to SINDy, ESINDy, and SPL. The recovered equations,

$$\dot{x} = -0.960\,y - 1.001\,z,$$
$$\dot{y} = 0.921\,x + 0.180\,y,$$
$$\dot{z} = 0.241 + 0.968\,xz - 5.615\,z,$$

are within $5\%$ of the ground truth across coefficients, including the decomposition of $z(x - c)$ into $xz$ and $-cz$.

## H  BALL DROP DYNAMICS WITH AIR RESISTANCE

The discovery of governing equations for falling objects represents a fundamental challenge extending from Galileo's idealized free-fall equation $H(t) = h_0 + v_0 t - \frac{1}{2} g t^2$ to real-world scenarios involving complex air resistance effects (Clancy, 1975; Greenwood et al., 1986). This case study demonstrates CaNDiCE's capability for discovering physics from experimental data using ball-drop trajectories from de Silva et al. (2020), encompassing eleven distinct ball types (baseball, blue basketball, green basketball, volleyball, bowling ball, golf ball, tennis ball, whiffle ball 1, whiffle ball 2, yellow whiffle ball, and orange whiffle ball) dropped from bridge height with 30 Hz sampling. The time between dropping and landing varies significantly across ball types due to diverse air resistance effects induced by differences in mass, surface texture, diameter, and aerodynamic properties. For the current analysis, 7 of these balls were considered (see Table 4)

We benchmark CaNDiCE against four established approaches. Three physics-based models serve as theoretical baselines (Table 3), with coefficients estimated via Powell's optimization (Powell,

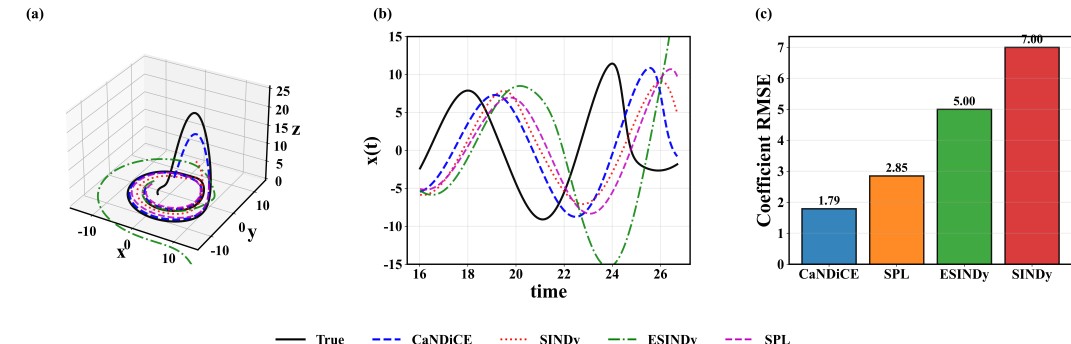

Figure 5: Rössler system recovery comparison: (a) 3D phase portraits, (b) time series (true vs. recovered), and (c) coefficient RMSE highlighting CaNDiCE's accuracy relative to SPL, ESINDy, and SINDy.

1964). The fourth baseline is the Symbolic Physics Learner (SPL) (Sun et al., 2022), representing state-of-the-art symbolic regression using Monte Carlo Tree Search.

Table 3: Physics-based baseline models for falling objects with air resistance.

| Model | Mathematical Expression |
|---|---|
| Model-1[a] | $H(t) = c_0 + c_1 t + c_2 t^2 + c_3 t^3$ |
| Model-2[b] | $H(t) = c_0 + c_1 t + c_2 e^{c_3 t}$ |
| Model-3[c] | $H(t) = c_0 + c_1 \log(\cosh(c_2 t))$ |

[a] https://faraday.physics.utoronto.ca/IYearLab/Intros/FreeFall/FreeFall.html
[b] https://physics.csuchico.edu/kagan/204A/lecturenotes/Section15.pdf
[c] https://en.wikipedia.org/wiki/Free_fall

For CaNDiCE, we intentionally restrict the world–model library to $\Theta(t) = \{1, t, t^2, t^3, t^4, \cos t, \sin t\}$ with $(\lambda_{\text{TC}}, \lambda_{\text{sp}}) = (10, 10)$ to probe failure modes when *causal* terms are absent. This basis omits functions that are physically relevant for drag, most notably $\log(\cosh(\cdot))$ and $\exp(\cdot)$, which appear repeatedly in the stronger baselines reported in Table 4. The consequence is a systematic model–form error: CaNDiCE must approximate dissipative effects with low–order polynomials (or trigonometric surrogates), yielding biased fits and reduced out–of–sample accuracy. Quantitatively, CaNDiCE is best on only *two* of the seven test objects (Baseball: RMSE = 0.420 vs. SPL 0.451; Golf ball: 0.090 vs. SPL 0.095), and is merely competitive but not best on Green Basketball (0.320 vs. SPL 0.297 and Model-1 0.316). Where the omitted causal terms are essential, performance degrades sharply: for **Volleyball** the learned linear model $H(t) = -0.1558 t$ cannot reproduce the curvature introduced by drag; CaNDiCE's error (RMSE = 10.68) is $\sim 32\times$ SPL (0.333). For **Whiffle Ball 1** and **Whiffle Ball 2**, the baselines exploit hyperbolic/exponential structure [e.g., $H(t) = 47.093 \, e^{-0.0788 t^2}$ and $H(t) = -18.606 + 65.858 \, e^{-0.0577 t^2}$], achieving 0.458 and 0.315 RMSE, respectively, while CaNDiCE—constrained to polynomials—settles at 0.600 and 0.340. A similar pattern holds for the **Tennis ball**, where the best model uses a nonpolynomial term ($\log(\cosh(\cdot))$, RMSE 0.464) and CaNDiCE trails at 0.500.

These results underscore a key limitation: *without the correct causal library elements,* the stochastic inverse/world-model machinery can neither propose nor validate the right mechanisms, and the counterfactual learner cannot recover them ex nihilo. In contrast to earlier case studies where the library contained the causal terms and CaNDiCE matched or exceeded the baselines, the ball-drop experiments reveal that (i) model-form error dominates when $\Theta$ lacks $\log(\cosh(\cdot))$ and $\exp(\cdot)$; (ii) the topological-causality regularizer $\lambda_{\text{TC}}$ preserves structure *within* the chosen family but cannot compensate for a misspecified family; and (iii) even modest misspecification can inflate error by an order of magnitude. Practically, CaNDiCE requires either (a) an expanded, physics-aware dictionary

Table 4: Physics discovery results for ball drop experiments comparing different methods

| Ball Type | Method & Expression | RMSE | DICE score |
|---|---|---|---|
| **Baseball** | CaNDiCE: $H(t) = 45.19 - 3.796\,t + 0.05894\,t^2$ | **0.420** | **1.0** |
| | SPL: $H(t) = -5.7707\,t^2 + 1.4959\,t + \log(\cosh(t^2)) + 47.6990$ | 0.451 | 0.8 |
| | SINDy: $H(t) = 43.488 - 3.682\,t + 0.057\,t^2 - 0.756 \sin -0.244 \cos t$ | 1.202 | 0.67 |
| | ESINDy: $H(t) = -48.413 + 0.942\,t - 0.666 \sin(t) - 0.956 \cos(t)$ | 1.02 | 0.4 |
| | WeakSINDy: $H(t) = -1136.733 + 83.251\,t - 2.053\,t^2 + 0.007\,t^3$ | 7.384 | 0.8 |
| | ODEFormer: $H(t) = 0.2419\,t - 22.5183 \sin(1.3900 + 0.0214\,t)$ | 7.34 | 0.5 |
| | Model-1: $H(t) = 47.682 + 1.456\,t - 5.629\,t^2 + 0.376\,t^3$ | 1.673 | 0.8 |
| | Model-2: $H(t) = 45.089 - 8.156\,t + 5.448 \exp(0)$ | 9.727 | 0.67 |
| | Model-3: $H(t) = 48.051 - 182.248 \log(\cosh(0.218\,t))$ | 1.823 | 0.0 |
| **Green Basketball** | CaNDiCE: $H(t) = -47.2 + 0.9517\,t$ | 0.320 | 0.67 |
| | SPL: $H(t) = -4.1465\,t^2 + \log(\cosh(1)) + 45.9087$ | **0.297** | 0.67 |
| | SINDy: $H(t) = -45.941 + 0.916\,t + 0.761 \sin(t) - 0.386 \cos t$ | 0.646 | 0.4 |
| | ESINDy: $H(t) = -45.667 + 0.910\,t + 0.727 \sin(t) - 0.363 \cos(t)$ | 0.610 | 0.67 |
| | WeakSINDy: $H(t) = -390.743 + 29.162\,t - 0.753\,t^2$ | 6.003 | **1.0** |
| | ODEFormer: $H(t) = 0.3392\,t - 25.3851 \sin(13.8000 + 0.0219\,t)$ | 7.46 | 0.5 |
| | Model-1: $H(t) = 46.438 - 0.340\,t - 3.882\,t^2 - 0.055\,t^3$ | 0.316 | 0.8 |
| | Model-2: $H(t) = 43.512 - 8.043\,t + 5.346 \exp(0)$ | 9.242 | 0.67 |
| | Model-3: $H(t) = 46.391 - 123.56 \log(\cosh(0.264\,t))$ | 1.265 | 0.0 |
| **Golf Ball** | CaNDiCE: $H(t) = 49.921 - 1.545\,t + 0.030\,t^2$ | **0.090** | **1.0** |
| | SPL: $H(t) = -4.9633\,t^2 + \log(\cosh(t)) + 49.5087$ | 0.095 | 0.5 |
| | SINDy: $H(t) = 27.472 - 2.869\,t + 0.046\,t^2 - 0.351 \sin +0.130 \cos t$ | 0.920 | 0.67 |
| | ESINDy: $H(t) = -50.337 + 0.945\,t - 0.679 \sin(t) + 0.252 \cos(t)$ | 0.529 | 0.4 |
| | WeakSINDy: $H(t) = -285.979 + 20.521\,t - 0.519\,t^2$ | 7.840 | **1.0** |
| | ODEFormer: $H(t) = 0.3392\,t - 27.0366 \sin(13.8000 + 0.0205\,t)$ | 7.56 | 0.5 |
| | Model-1: $H(t) = 49.413 + 0.532\,t - 5.061\,t^2 + 0.102\,t^3$ | 0.463 | 0.8 |
| | Model-2: $H(t) = 46.356 - 8.918\,t + 5.964 \exp(0)$ | 9.281 | 0.67 |
| | Model-3: $H(t) = 49.585 - 178.47 \log(\cosh(0.230\,t))$ | 1.298 | 0.0 |
| **Volleyball** | CaNDiCE: $H(t) = -0.1558\,t$ | 10.68 | 0.67 |
| | SPL: $H(t) = 48.0744 - 3.7772\,t^2$ | **0.333** | 0.67 |
| | SINDy: $H(t) = 52.814 - 3.966\,t + 0.059\,t^2 - 0.636 \sin +0.196 \cos t$ | 10.566 | 0.67 |
| | ESINDy: $H(t) = -47.414 + 0.921\,t - 0.983 \sin(t) - 0.587 \cos(t)$ | 1.027 | 0.4 |
| | WeakSINDy: $H(t) = -701.017 + 50.613\,t - 1.239\,t^2$ | 5.259 | **1.0** |
| | ODEFormer: $H(t) = -0.1814\,t$ | 5.83 | 0.67 |
| | Model-1: $H(t) = 48.046 + 0.362\,t - 4.352\,t^2 + 0.218\,t^3$ | 0.758 | 0.8 |
| | Model-2: $H(t) = 45.320 - 7.317\,t + 5.037 \exp(0)$ | 9.000 | 0.67 |
| | Model-3: $H(t) = 48.123 - 107.06 \log(\cosh(0.271\,t))$ | 0.876 | 0.0 |
| **Tennis Ball** | CaNDiCE: $H(t) = 80.19 - 5.317\,t + 0.07548\,t$ | 0.500 | 0.67 |
| | SPL: $H(t) = -3.9859\,t^2 + \log(\cosh(1)) + 47.3888$ | 0.828 | 0.67 |
| | SINDy: $H(t) = 72.217 - 4.926\,t + 0.071\,t^2 - 0.650 \sin -0.126 \cos t$ | 1.009 | 0.67 |
| | ESINDy: $H(t) = -45.598 + 0.884\,t - 0.640 \sin(t) - 0.773 \cos(t)$ | 0.499 | 0.4 |
| | WeakSINDy: $H(t) = -331.091 + 25.019\,t - 0.655\,t^2$ | 6.991 | **1.0** |
| | ODEFormer: $H(t) = 4.6512\,t\,(-0.1275 + 0.0021\,t)$ | 5.87 | **1.0** |
| | Model-1: $H(t) = 47.738 + 0.658\,t - 4.901\,t^2 + 0.325\,t^3$ | 0.496 | 0.8 |
| | Model-2: $H(t) = 45.016 - 7.717\,t + 5.212 \exp(0)$ | 8.501 | 0.67 |
| | Model-3: $H(t) = 47.874 - 113.108 \log(\cosh(0.270\,t))$ | **0.464** | 0.0 |
| **Whiffle Ball 1** | CaNDiCE: $H(t) = 109.7 - 6.523\,t + 0.08847\,t^2$ | 0.600 | **1.0** |
| | SPL: $H(t) = 47.093 \exp(-0.0788\,t^2)$ | 2.302 | 0.0 |
| | SINDy: $H(t) = 91.102 - 5.592\,t + 0.077\,t^2 + 0.151 \sin -0.492 \cos t$ | 0.821 | 0.67 |
| | ESINDy: $H(t) = -39.928 + 0.787\,t + 0.081 \sin(t) - 1.045 \cos(t)$ | 0.444 | 0.4 |
| | WeakSINDy: $H(t) = -1162.424 + 85.451\,t - 2.109\,t^2 + 0.008\,t^3$ | 9.037 | 0.8 |
| | ODEFormer: $H(t) = -0.1417\,t$ | 5.83 | 0.67 |
| | Model-1: $H(t) = 46.969 + 0.574\,t - 4.505\,t^2 + 0.522\,t^3$ | 1.379 | 0.8 |
| | Model-2: $H(t) = 44.259 - 6.373\,t + 4.689 \exp(0)$ | 8.089 | 0.67 |
| | Model-3: $H(t) = 47.062 - 34.083 \log(\cosh(0.462\,t))$ | **0.458** | 0.0 |
| **Whiffle Ball 2** | CaNDiCE: $H(t) = 101.1 - 6.138\,t + 0.08393\,t^2$ | 0.340 | **1.0** |
| | SPL: $H(t) = -18.6063 + 65.8583 \exp(-0.0577\,t^2)$ | **0.315** | 0.0 |
| | SINDy: $H(t) = 98.323 - 5.998\,t + 0.082\,t^2 - 0.305 \sin -0.247 \cos t$ | 1.515 | 0.67 |
| | ESINDy: $H(t) = -41.242 + 0.809\,t - 0.291 \sin(t) - 1.001 \cos(t)$ | 0.480 | 0.4 |
| | WeakSINDy: $H(t) = -625.046 + 46.212\,t - 1.158\,t^2$ | 5.047 | **1.0** |
| | ODEFormer: $H(t) = -0.1552\,t$ | 5.83 | 0.67 |
| | Model-1: $H(t) = 47.215 + 0.296\,t - 4.379\,t^2 + 0.421\,t^3$ | 0.793 | 0.8 |
| | Model-2: $H(t) = 44.443 - 6.744\,t + 4.813 \exp(0)$ | 7.651 | 0.67 |
| | Model-3: $H(t) = 47.255 - 38.29 \log(\cosh(0.447\,t))$ | 0.983 | 0.0 |

that includes canonical drag surrogates, or (b) a meta-library mechanism that proposes expressive candidates (e.g., hyperbolic/exponential atoms) and tests them under the same causal criteria. Ab-

sent such terms, the method remains biased toward polynomial surrogates and fails to attain the performance seen in prior studies where the causal structure was representable by the library.

## I  ABLATION AND SENSITIVITY ANALYSIS

To understand the contribution of individual mechanistic terms to causal identifiability, we conduct an ablation-style sensitivity analysis on the recovered governing equations for Lorenz system across 25 different runs. Rather than perturbing coefficients, we progressively remove basis functions from the learned model—starting with the least influential terms ranked by their counterfactual MSE—and recompute the resulting TC. This produces a deletion trajectory that reveals how robust the causal structure is to model sparsification. Figure 6 shows the variation in the Dice equation similarity, scaled RMSE, and the change in TC as physics terms are incrementally removed from the recovered equations.

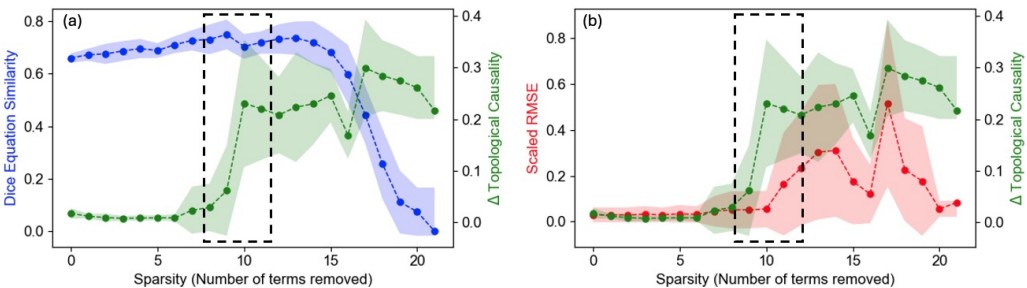

Figure 6: (a) Dice equation similarity and the change in TC as physics terms are incrementally removed, making the inferred governing equations sparse and (b) shows the scaled RMSE and the change in TC as physics terms are incrementally removed.

Interestingly, change in TC (as observed in constraint in Eq. 10) remains relatively stable as non-essential terms are removed, but exhibits a sharp change once a truly causal interaction term is ablated, indicating a phase-transition–like sensitivity to structural perturbations. This analysis highlights which terms are structurally indispensable for preserving the system's causal geometry and provides an interpretable diagnostic complementary to coefficient error metrics.

Additionally, from Figure 6(b), we notice that if the TC is ignored, and only RMSE is used, it is not possible to distinguish between solutions with no sparsity (e.g., when no terms are removed sparsity = 0) vs when almost all the terms are removed (for instance, at sparsity of 20). As such, a counterfactual explanation approach that relies solely on sparsity, i.e., perturbing the minimal number of terms, is only informed by the goodness of fit. Hence, it cannot distinguish between terms that merely improve predictive accuracy and those that encode essential causal interactions. In such a setting, small-magnitude but structurally critical coefficients may be removed without any immediate penalty, leading to counterfactuals that remain accurate in short-term prediction yet fundamentally violate the underlying causal geometry of the system.

