# OpenReview forum: "CaNDiCE: Causal Discovery of Nonlinear Dynamics Through Counterfactual Explanations"
_ICLR.cc/2026/Conference — Submitted to ICLR 2026_

### Official Review · Reviewer_H6Lp · 2025-10-24

**Soundness:** 3
**Presentation:** 2
**Contribution:** 3
**Rating:** 4
**Confidence:** 3

**Summary:**

The paper proposes an equation discovery framework for dynamical systems based on topological causality. It first defines a world model as the posterior of the sparse regression parameters given the bootstrapped data samples and updates the posterior using a stochastic inverse approach. Then, it trains a generator that generates perturbations to the sampled parameters from the world model using a combination of GAN loss and several regularization terms that promote sparsity and causal consistency of the perturbations. The terms whose coefficients are often perturbed by the trained generator are deemed causal and retained in the function library for sparse regression.

**Strengths:**

* The paper provides extensive context about related topics, such as topological causality and counterfactuals.
* The experimental results are impressive on the Lotka-Volterra equation and the Lorenz equation. The proposed method did very well when limited data were available.

**Weaknesses:**

* I find the writing in the methodology section unclear in general. For example, in Section 3.1, the kernel density estimator and stochastic inverse approach are crucial for estimating and updating the posterior $\pi(\Theta\Lambda|\mathbf y)$. These techniques should be (at least briefly) introduced in this problem context. Also, the notation $\pi(\Theta\Lambda|\mathbf y)$ is slightly confusing. I suppose $\Theta\Lambda$ here in fact refers to the simulated trajectory from $\dot {\mathbf x} = \Theta\Lambda$, but this notation alone might suggest the RHS expression itself.
* The presentational issues become more severe when it comes to the counterfactual model (Section 3.2 and 3.3). I have a lot of questions regarding these sections. In the CGAN formulation, what is the difference between the role of the discriminator and the classifier? They seem to both classify between the unperturbed (real) parameters and the perturbed (fake) parameters. Also, from the algorithm, it seems that the update of D is decoupled from that of G, which is different from the original GAN formulation. Why is that? I cannot understand eq. (10) either. The classification label $z$ is never explained or defined in the text. And the $\mathbf y$ inside the expectation does not make sense, since this is a scalar equation.
* The writing should be improved in general. For example, the second point of the contribution list says that "... generating counterfactual instances that lead to *out-of-distribution* trajectories... Counterfactuals are obtained by... *in-distribution* trajectories." Before reading the later sections, I could not figure out whether the counterfactual model should lead to in-distribution or out-of-distribution trajectories. On second thought, I can understand this statement where "minimally perturbing" seems to serve as a negative, but the unnecessary complexity in writing has made the paper difficult to understand.
* It would be great to include a figure explaining the entire pipeline, in addition to Algorithm 1.
* The experiments only considered two simple dynamical systems. While the results on these two systems are impressive, it remains to be seen whether this can be generalized to other datasets.
* The experiments did not compare with weak SINDy, which is specifically designed for noisy data.

**Questions:**

* L60: missing cross-reference
* distinguish between \citet and \citep
* L219: Can you elaborate on why you need the causal consistency constraint?
* How many samples are needed to train the CGAN counterfactual model? Since it involves some neural networks, does it do well with the small sample sizes in the experiments?
* How time-efficient is the proposed method compared to baseline SINDy?

---

> ### Author Response · Authors · 2025-11-23
>
> We thank the reviewer for their constructive feedback. Please see response to your concerns.
>
> 1. I find the writing in the methodology section unclear in general. For example, in Section 3.1, the kernel density estimator and stochastic inverse approach are crucial for estimating and updating the posterior π(ΘΛ|y). These techniques should be (at least briefly) introduced in this problem context. Also, the notation is slightly confusing. I suppose ΘΛ here in fact refers to the simulated trajectory from x = ΘΛ, but this notation alone might suggest the RHS expression itself.
>
> Response: We thank the reviewer for their comments. We have provided further details about constructing the world model using the stochastic inverse approach and kernel density estimator at the beginning of Section 3. Due to page limitations, we have moved the discussion on bootstrapping to Appendix B.
>
> 2. The presentational issues become more severe when it comes to the counterfactual model (Section 3.2 and 3.3). I have a lot of questions regarding these sections. In the CGAN formulation, what is the difference between the role of the discriminator and the classifier? They seem to both classify between the unperturbed (real) parameters and the perturbed (fake) parameters.
>
> Response. The discriminator is testing whether the proposed perturbations of the parameter are realistic, i.e., the likelihood of being sampled from the world model. In contrast, the classifier is testing whether the perturbed coefficients will lead to an out-of-distribution trajectory. The feedback from the classifier is used to train the generator.
>
> 3. Also, from the algorithm, it seems that the update of D is decoupled from that of G, which is different from the original GAN formulation. Why is that? I cannot understand eq. (10) either.
>
> Response. We thank the reviewer for pointing this out. The discriminator and the generator are simultaneously trained as indicated in Eq (8). This was a typo, and we have updated this in the algorithm.
>
> 4. The classification label is never explained or defined in the text. And the \bm{y} inside the expectation does not make sense, since this is a scalar equation.
>
> Response. The classifier is based on a simple KS statistic as mentioned in Eq. (7) with a significance level of 0.05. Please see the last paragraph of Section 3.1. We acknowledge the oversight in the formula for the cross-entropy loss. \bm{y} is in fact simply z, the label for when the perturbed coefficient leads to out of distribution trajectory.
>
> 5. The writing should be improved in general. For example, the second point of the contribution list says that "...generating counterfactual instances that lead to out-of-distribution trajectories... Counterfactuals are obtained by...in-distribution trajectories." Before reading the later sections, I could not figure out whether the counterfactual model should lead to in-distribution or out-of-distribution trajectories. On second thought, I can understand this statement where "minimally perturbing" seems to serve as a negative, but the unnecessary complexity in writing has made the paper difficult to understand.
>
> Response. We thank the reviewer for their feedback. We have simplified the language to make it clear.
>
> 6. It would be great to include a figure explaining the entire pipeline, in addition to Algorithm 1.
>
> Response. Yes, we have added Figure 1 to show high-level overview of the algorithm.
>
> 7. The experiments only considered two simple dynamical systems. While the results on these two systems are impressive, it remains to be seen whether this can be generalized to other datasets.
>
> Response. Please note that the appendix section already contained an extensive analysis of the approach in the presence of noise with 3 additional systems, along with a real-world experiment on ball dropping. However, to make the results more exhaustive and demonstrate generalizability, we have evaluated 10 additional systems (with 3 and 4 dimensions) taken from the ODEBench dataset. Note that these are the largest problems in the dataset. The new results can be found in Table 1. We are still generating the results for SPL and will update in the next few days.  We have also included two additional state-of-the-art methods pointed out by other reviewers, namely weak SINDy and ODEFormer.
>
> 8. The experiments did not compare with weak SINDy, which is specifically designed for noisy data.
>
> Response. This is now included in the revised manuscript in Table 1.

---

> > ### Author Response · Authors · 2025-11-23
> >
> > Please see response to the additional questions below --
> > 1. L60: missing cross-reference
> >
> > Response: This is an oversight; we have included the correct reference on this line
> >
> > 2. distinguish between \citet and \citep
> >
> > Response: Thanks for pointing this. We have corrected the inconsistency in citing.
> >
> > 3. L219: Can you elaborate on why you need the causal consistency constraint?
> >
> > Response. Through the causal consistency constraint, we encourage those perturbations that minimally alter the causal structure of the underlying governing equations as discussed in Section 3.2, lines 248-251. The importance of causal consistency is further discussed in Proposition 1 from a causal identifiability perspective in Section 3.4.
> >
> > 4. How many samples are needed to train the CGAN counterfactual model? Since it involves some neural networks, does it do well with the small sample sizes in the experiments?
> >
> > Response. All of our experiments are tested on trajectories with 1000 time stamps. The results with varying sample sizes are presented in Figure 2.
> >
> > 5. How time-efficient is the proposed method compared to baseline SINDy?
> >
> > This remains one of the limitations of our approach. The time complexity is now included in Table 1. However, the longest time it took was 2100 seconds (~35 mins). The column regarding SPL is empty as we are still generating the results for it and will update in the next few days.

---

> > > ### Author Response · Authors · 2025-11-27
> > > **Updates on remaining results**
> > >
> > > Dear reviewer,
> > >
> > > We have now completed the remaining comparisons and updated the manuscripts with the results on SPL in Table 1, which can be found on page 10. Thank you again for your constructive feedback. Please let us know if you have any additional questions/concerns.

---

> > > > ### Comment · Area_Chair_VmUd · 2025-11-28
> > > >
> > > > Dear Reviewer,
> > > >
> > > > Please make sure you read the authors' response and engage with them in the discussion before the end of the discussion period on **Dec 03 '25 09:00 PM UTC**. This is a hard deadline.
> > > >
> > > > Thank you for supporting quality peer review at ICLR.
> > > >
> > > > AC

---

### Official Review · Reviewer_hwVS · 2025-10-31

**Soundness:** 3
**Presentation:** 3
**Contribution:** 3
**Rating:** 4
**Confidence:** 4

**Summary:**

This paper addresses the problem of equation discovery from noisy and limited data, which is a long-standing challenge in scientific machine learning. The authors introduce CaNDiCE (Causal Discovery of Nonlinear Dynamics through Counterfactual Explanations), a framework that integrates counterfactual reasoning and topological causality to recover parsimonious, causally meaningful governing equations.
The method builds a world model over the coefficients of a predefined basis library, generates counterfactual coefficients via a conditional GAN that satisfy sparsity and causal consistency constraints, and then identifies the minimal causal set of terms to refit a sparse regression model.
Empirical evaluations span five benchmark dynamical systems: Lotka–Volterra, Lorenz, Van der Pol, Rössler, and a ball-drop experiment with air resistance, under varying signal-to-noise ratios (SNR) and data regimes. Results show large improvements compared to classical and recent baselines such as SINDy, ESINDy, and SPL, especially in low-data and high-noise conditions.

**Strengths:**

1. **Conceptual novelty.** The paper introduces a fresh causal perspective on equation discovery by embedding counterfactual generation within a causal consistency regularization. This adversarial counterfactual setup, where coefficient perturbations are constrained by topological causality, is an original and very interesting idea.

2. **New inspiration for physics discovery using causal constraints.** The incorporation of topological causality (via cross-mapping on reconstructed manifolds) provides a principled way to ensure causal consistency in dynamical systems where classical DAG-based causality notions fail. This is a meaningful step toward causal interpretability in scientific ML.

3. **Clear writing and methodological exposition.** The paper is well structured and mathematically precise. Algorithm 1 and the breakdown into “world model”, “counterfactual model”, and “minimum set discovery” make the pipeline understandable.

**Weaknesses:**

1. **Missing key baselines and contextualization.** The paper omits several recent transformer- or diffusion-based approaches to symbolic regression and equation discovery, notably "ODEFormer: Symbolic Regression of Dynamical Systems with Transformers (ICLR 2024)", which introduces the ODE-Bench dataset. Including such models would better position CaNDiCE within the current landscape of neural-symbolic discovery.

2. **Theoretical grounding and identifiability.** While the paper discusses causal constraints qualitatively, it lacks a formal analysis of parameter identifiability or conditions under which the causal coefficients are recoverable. For example, under what assumptions does topological causality regularization guarantee recovery of the correct sparse structure?

3. **Computational complexity and scalability.**
The combination of stochastic inversion, bootstrapping, and GAN training raises concerns about computational efficiency. The paper would benefit from a runtime or memory comparison with baselines, especially for higher-dimensional systems.

4. **Interpretability and intuition gaps.**
The notion of topological causality may be unfamiliar to much of the ICLR audience. A concise **visual** example—e.g., a 2-variable dynamical system with the corresponding manifold mappings—would greatly help build intuition.

5. **Library dependence and limitations.**
As shown in the ball-drop case study, CaNDiCE’s success depends heavily on whether the causal functional forms are representable within the predefined basis library. The paper should discuss possible extensions to mitigate this limitation.

**Questions:**

1. **Identifiability and guarantees**
Under what assumptions does CaNDiCE provably recover the correct causal terms? Is there an identifiable mapping between topological-causality preservation and coefficient consistency?

2. **Complexity and scalability**
What is the asymptotic or empirical computational cost relative to SINDy/ESINDy/SPL? Could the GAN-based counterfactual generation become a bottleneck for high-dimensional systems?

3. **Robustness to library misspecification**
Can the model adaptively extend or refine its basis set when key functional forms are missing?

4. **Ablation or sensitivity analysis**
How sensitive are results to the hyperparameters λ_TC and λ_sp? Does the balance between sparsity and causality penalties substantially affect which terms are identified as causal?


I am happy to raise my score if the concerns in questions and weaknesses are addressed.

---

> ### Author Response · Authors · 2025-11-23
>
> We are grateful to the reviewers for the constructive feedback. Please see our response to the weaknesses and questions.
>
> Weaknesses:
> 1. Missing key baselines and contextualization. The paper omits several recent transformer- or diffusion-based approaches to symbolic regression and equation discovery, notably "ODEFormer: Symbolic Regression of Dynamical Systems with Transformers (ICLR 2024)", which introduces the ODE-Bench dataset. Including such models would better position CaNDiCE within the current landscape of neural-symbolic discovery.
>
> Response. We are grateful for referring us to ODEFormer. We have included this and related papers in our introduction and also compared our method with the three and four-dimensional systems in the ODEBench dataset, and are now in the revised paper (see Table 1). Note that the SPL columns say TBA as we are still generating the results for SPL and will update them in the next few days.  As you can see, ODEFormer, being the state-of-the-art method, focuses on fitting the data and does not care about the actual functional form, which is evident from the proposed DiCE score. In contrast, our counterfactual approach leads to significantly better performance (nearly 48% higher as compared to ODEFormer)
>
> 2. Theoretical grounding and identifiability. While the paper discusses causal constraints qualitatively, it lacks a formal analysis of parameter identifiability or conditions under which the causal coefficients are recoverable. For example, under what assumptions does topological causality regularization guarantee recovery of the correct sparse structure?
>
> Response. We have provided a theoretical perspective on the identifiability of the causal structure under the topological causality regularization. The proposition argues that the topological causality is unique to a given dynamical system and that the addition/removal of a physics term non-trivially alters the causal structure and therefore the topological causality. Please see Section 3.4.
>
> 3. Computational complexity and scalability. The combination of stochastic inversion, bootstrapping, and GAN training raises concerns about computational efficiency. The paper would benefit from a runtime or memory comparison with baselines, especially for higher-dimensional systems.
>
> Response. Certainly, our method requires more time to perform the construction of the world model, bootstrapping, and GAN training (which also involves the computation of topological causality). For the new results on higher-dimensional problems, we have provided the run times against weak SINDY and ODEFormer in Table 1. Note that the SPL columns say TBA as we are still generating the results for SPL and will update them in the next few days.
>
> 4. Interpretability and intuition gaps. The notion of topological causality may be unfamiliar to much of the ICLR audience. A concise
> Visual example—e.g., a 2-variable dynamical system with the corresponding manifold mappings—would greatly help build intuition.
>
> Response. We thank the reviewer for this suggestion. Due to page limitations, we have added a simple example of topological causality in Appendix A.
>
> 5. Library dependence and limitations. As shown in the ball-drop case study, CaNDiCE’s success depends heavily on whether the causal functional forms are representable within the predefined basis library. The paper should discuss possible extensions to mitigate this limitation.
>
> Response. We thank the reviewer for their comment. Yes, we have included some ideas for future research where we could actively update the library if the current library does not perform well or capture the variability of the system.

---

> ### Author Response · Authors · 2025-11-23
>
> Please see response to the questions raised (similar to weakenesses)
> Questions:
> 1. Identifiability and guarantees. Under what assumptions does CaNDiCE provably recover the correct causal terms?Is there an identifiable mapping between topological-causality preservation and coefficient consistency?
>
> Response: The manuscript has been updated to include discussion on the identifiability of the CANDiCE method in Section 3.4.
>
> 2. Complexity and scalability: What is the asymptotic or empirical computational cost relative to SINDy/ESINDy/SPL? Could the GAN-based counterfactual generation become a bottleneck for high-dimensional systems?
>
> Response. Certainly, computational cost is one of the limitations of our approach and has been summarized in Table 1.  Clearly, this is one of the limitations of our approach. However, please note that even the slowest example completes within 35 minutes on the CPU.
>
> 3. Robustness to library misspecification Can the model adaptively extend or refine its basis set when key functional forms are missing?
>
> Response: Currently, the method does not have any mechanism to adaptively refine and extend the predefined physics library. This is why one of the key assumptions we have made is that the predefined library is an overcomplete library containing all the necessary terms in the governing equation and some spurious ones. We understand that this assumption might not always hold true in several real world systems and we have planned to address this issue as part of the future work on this method using active learning methods. We have included this in the Conclusion section of the manuscript.
>
> 4. Ablation or sensitivity analysis How sensitive are results to the hyperparameters λ_TC and λ_sp? Does the balance between sparsity and causality penalties substantially affect which terms are identified as causal?
>
> Response: We are still working on the sensitivity analysis and will update the results in the next few days.

---

> > ### Comment · Reviewer_hwVS · 2025-11-25
> >
> > I thank the authors for their effort. While some results are still incomplete (e.g., the comparison with SPL on the ODE Bench and the sensitivity analysis), I am pleased to see the substantial improvements in the revised manuscript, including the discussion on the theoretical identifiability of topological causality, the additional results and references related to ODEFormer, as well as the new computational cost analysis.
> >
> > I also find the proposed DICE score to be a valuable addition for evaluating equation discovery algorithms. It would be great to formally introduce this metric in the paper (and perhaps briefly check or discuss whether related metrics exist—I am not fully familiar with that area) and to report it for the other experiments as well.
> >
> > Overall, the updates have addressed most of my previous concerns. Please let me know once the missing results are included, and I will be happy to increase my score.

---

> > > ### Author Response · Authors · 2025-11-27
> > >
> > > Dear reviewer,
> > >
> > > Thank you again for your constructive feedback and positive remarks. These have been very helpful in improving the quality of the paper. We have now completed the remaining comparisons and sensitivity analysis. We updated the manuscript with the results on SPL in Table 1, which can be found on page 10, and the ablation study is presented in Appendix I. Thank you again for your constructive feedback. Please let us know if you have any additional questions/concerns.

---

### Official Review · Reviewer_xPLj · 2025-10-31

**Soundness:** 3
**Presentation:** 3
**Contribution:** 2
**Rating:** 2
**Confidence:** 5

**Summary:**

The authors introduce a couterfactional penalty term to model discovery methods, specifically the spare identification of nonlinear dynamics algorithm.  They then show the performance of this method against some of the variants of SINDy.

**Strengths:**

The ideas of the paper are actually quite nice.  It seems a rather nice innovation to include in the regression architecture.  I would strongly encourage the authors to continue pursuing this line of work as it has great potential.

**Weaknesses:**

Unfortunately, the method seems rathe immature to me at this point.  Specifically, the two models demonstrated in the paper are the Lotka-Volterra and Lorenz system, both of which are very basic models.  It certainly fine to demonstrate initially on these models, but it certainly would be expected for an ICLR to have much more challenging models to explore.

Additionally, the comparisons to SINDy, eSINDy, SPL, while good, are certainly not state-of-the-art methods.  In fact, these methods are not really aimed at causal inference.  Much more serious comparisons should be made against what are considered causal learning methods.  So the comparisons are simply not up to what would be expected.

**Questions:**

Only two main questions:

How does this work for more challenging models than Lorenz/Lotka-Volterra?  Especially with noise?

How does the method actually hold up in comparison with leading causality inference methods?

---

> ### Author Response · Authors · 2025-11-23
>
> We appreciate the constructive feedback from the reviewer. Please see our response to your questions below.
>
> 1. How does this work for more challenging models than Lorenz/Lotka-Volterra? Especially with noise?
>
> Response. We would like to highlight that,
> (a) The appendix section already contained an extensive analysis of the approach in the presence of noise with 3 additional systems, along with a real-world experiment on ball dropping. However, to make the results more exhaustive and demonstrate generalizability, we have evaluated 10 additional systems (with 3 and 4 dimensions) taken from the ODEBench dataset. Note that these are the largest problems in the dataset. The new results can be found in Table 1. We are still generating the results for SPL and will update in the next few days.
>
> (b) We have also included two additional state-of-the-art methods pointed out by other reviewers, namely weak SINDy and ODEFormer
>
> 2. How does the method actually hold up in comparison with leading causality inference methods?
>
> Response. Regarding comparison with causal inference techniques, please note that traditional causal inference techniques are not applicable to the case of discovering governing equations (already discussed in the paper introduction, page 2, lines 63 onwards). Specifically, dynamical systems exhibit non-separability, where each observable contains, in principle, sufficient information about the full system state. Hence, traditional methods are not directly applicable. This challenge motivated the concept of topological causality, which we have integrated as part of our approach.
>
> Nevertheless, we did find a few recent papers that have explored causality in the discovery of governing equations, e.g., [A]. However, the paper assumed that the functional form is known and is only concerned with estimating the coefficients. Other works, e.g., [B], have only indirectly investigated the causality of the recovered equations.
>
> [A] - Yao, D., Muller, C., & Locatello, F. (2024). Marrying causal representation learning with dynamical systems for science. Advances in Neural Information Processing Systems, 37, 71705-71736.
> [B] - O'Brien, A. (2024). Dynamic Causality: Sparse Symbolic Regression as a Tool to Learn Causal Dynamic Structural Equations with Applications to Counterfactuals (Doctoral dissertation, Drexel University).
>
> We apologize in advance if we missed any specific literature that you might be referring to, and we would be glad to review it.

---

> > ### Author Response · Authors · 2025-11-27
> > **Updates on remaining results**
> >
> > Dear reviewer,
> >
> > We have now completed the remaining comparisons and updated the manuscripts with the results on SPL in Table 1, which can be found on page 10. Thank you again for your constructive feedback. Please let us know if you have any additional questions/concerns.

---

> > > ### Comment · Area_Chair_VmUd · 2025-11-28
> > >
> > > Dear Reviewer,
> > >
> > > Please make sure you read the authors' response and engage with them in the discussion before the end of the discussion period on **Dec 03 '25 09:00 PM UTC**. This is a hard deadline.
> > >
> > > Thank you for supporting quality peer review at ICLR.
> > >
> > > AC

---

### Official Review · Reviewer_CSxc · 2025-11-01

**Soundness:** 4
**Presentation:** 4
**Contribution:** 3
**Rating:** 6
**Confidence:** 4

**Summary:**

This work proposes a new approach to select relevant basis functions for sparse identification of governing equations based on topological causality. The authors propose to train a GAN to sample counterfactual trajectories that preserve the causal structure as measured by a topological causality metric. Examining the distribution of sampled counterfactuals then allows for the construction of a minimal library of basis functions by eliminating terms that do not generate causally consistent counterfactuals.

**Strengths:**

This paper is well-written and uses a recently developed theoretical tool for causal analysis of dynamical systems to improve system identification. The experiments show that this approach has a significant advantage over standard non-causal system identification, especially for noisy data where standard SINDy fails.

**Weaknesses:**

My main concern is regarding the scalability of the counterfactual generation. It seems to be at least quadratic in the number of initial library terms to even evaluate the loss for the GAN. Furthermore, the GAN must effectively sample all sparse combinations to truly find all good counterfactuals. As mentioned in the paper, this is NP-hard. The world model construction may also run into scalability issues due to the need to estimate a stochastic inverse.

**Questions:**

1. How computationally expensive is constructing the world model and computing the topological causality metric? How does it scale?
2. How stable and reproducible is the GAN training, and are there failure modes that you observe? Does failure to sample enough counterfactuals result in a library that is too small to fully capture the dynamics?
3. Can you run a test on a much higher-dimensional system to give a sense of the scaling behavior?
4. I'm a bit confused by the diversity-promoting loss in equation 11. If gamma > 0, wouldn't the loss encourage the new counterfactual candidate to be similar to pre-existing counterfactuals (so reducing diversity)? Also, what does it mean to subtract a coefficient from a set of coefficients (are you averaging here)?

---

> ### Author Response · Authors · 2025-11-23
>
> We thank the reviewers for their constructive feedback. Please see below our response --
>
> 1.  	How computationally expensive is constructing the world model and computing the topological causality metric? How does it scale?
>
> Response. Topological Causality (TC) has roughly O(T × k × d³) complexity per pair of variables, where T = number of time points (embeddings), k = number of nearest neighbors (typically 10), d = embedding dimension (usually 3-5). Finding nearest neighbors involves O(T × d × log T) (using k-NN search). Local linear regression and SVD for each point → O(T × k × d³) total (dominant term).  So for n variables, computing all directed pairs costs O(n² T k d³). For a 4-dimensional system (maximum in ODEBench) on Apple M1 Max chip, this amounts to a wall clock time of 0.00000215 seconds, and for the most complex system with 24 terms (complexity defined in terms of the number of basis functions, the wall clock time is 0.32486606 seconds.
>
> Constructing the world model, in contrast, requires MCMC sampling, which required an average of 517 seconds, with a minimum being 236 and a maximum of 1287 seconds. As such, the entire model run time (including MCMC for world model generation and Conditional GAN training) has been summarized in Table 1 of the revised paper. Note that the SPL columns say TBA as we are still generating the results for SPL and will update them in the next few days. Clearly, this is one of the limitations of our approach. However, please note that even the slowest example completes within 35 minutes on the CPU.
>
> 2.  	How stable and reproducible is the GAN training, and are there failure modes that you observe? Does failure to sample enough counterfactuals result in a library that is too small to fully capture the dynamics?
> Response. The library generation is conducted before the GAN training, which occurs during the world model construction. Once the library is generated, the GAN module only attempts to modify the coefficients. The only failure mode observed during GAN training is when the sampled coefficients lead to unstable trajectories, in which case we skip that coefficient. However, such instances belong to the tail of the posterior distribution in the world model, hence their frequency of occurrence is low (<1%).
>
> 3.  	Can you run a test on a much higher-dimensional system to give a sense of the scaling behavior?
> Response. Yes, we have conducted additional experiments following the three and four-dimensional problems presented in the ODEBench Dataset. The complexity of such systems is around 10 in terms of the number of basis terms.
>
> 4.  	I'm a bit confused by the diversity-promoting loss in equation 11. If gamma > 0, wouldn't the loss encourage the new counterfactual candidate to be similar to pre-existing counterfactuals (so reducing diversity)? Also, what does it mean to subtract a coefficient from a set of coefficients (are you averaging here)?
>
> Response. Thank you for pointing out the typo. There should be a minus sign. We have corrected that. Also, we are taking the average of the difference from the existing coefficient and have updated the equation to reflect this correction.

---

> > ### Author Response · Authors · 2025-11-27
> > **Updates on remaining results**
> >
> > Dear reviewer,
> >
> > We have now completed the remaining comparisons and updated the manuscripts with the results on SPL in Table 1, which can be found on page 10. Thank you again for your constructive feedback. Please let us know if you have any additional questions/concerns.

---

> > > ### Comment · Reviewer_CSxc · 2025-11-27
> > >
> > > Thank you for the response.
> > >
> > > Regarding scaling, 10 basis terms is arguably still rather small. If you had 100 terms, would it significantly hamper the performance due to difficulties with the world model sampling and GAN training?
> > >
> > > Furthermore, 3 or 4-dimensional systems are also relatively small. Why not consider something like Lorenz 96 or a large system of sparsely coupled oscillators?

---

> > > > ### Author Response · Authors · 2025-11-30
> > > >
> > > > Dear reviewer,
> > > >
> > > > Thank you again for your thoughtful feedback.
> > > >
> > > > 1. Scaling in the number of basis terms. While GAN training and sampling are computationally non-trivial, in our experience, they are not the primary bottleneck. Instead, the limiting step is the construction of the world model itself. As the dimensionality grows, the number of candidate features grows combinatorially, and the initial library-based world model becomes increasingly dense even before counterfactual refinement. To show this, we conducted additional experiments inspired by your suggestion. We applied our method to the Lorenz–96 system with 20 dimensions, where a second-order library leads to 231 candidate terms. The learned world model contained <100 active terms, and our GAN-based refinement was able to remove most spurious components, achieving a Dice Equation Similarity score of 0.96.
> > > >
> > > > 2. Scaling to very large systems. At 40 dimensions, however, the situation changes dramatically: a second-order library yields 861 features, and the resulting world model contains over 600 active terms. In this regime, GAN refinement becomes significantly more challenging—not because the GAN cannot sample counterfactuals, but because the initial model is already extremely dense, making it difficult to meaningfully navigate the perturbation space. As a result, performance degrades.
> > > >
> > > > 3. Scope and contribution of our method. Our primary contribution is not world-model construction itself (which remains limited by current sparse-regression techniques), but rather the use of topological causality as an additional causal signature that suppresses spurious terms. Even if the world model is imperfect, causality-based constraints allow us to remove non-causal components in a principled manner.
> > > >
> > > > Lastly, scaling world-model discovery to very large systems (e.g., 40+ dimensions) remains a broader open challenge in the literature—affecting Weak SINDy, UQ-SINDy, and related methods alike—and we view our work as complementary to upcoming advances in scalable sparse regression.

---

### Author Response · Authors · 2025-12-03
**Summary of revisions**

Dear reviewers and Area Chairs,
Thank you for the insightful feedback and the opportunity to respond and revise the manuscript. Below, we provide a summary of the changes made during the rebuttal phase for your perusal.

1. Methodology

Reviewer concern. Section 3 lacked clarity (world model construction, classifier vs. discriminator, notation). Topological Causality (TC) may be unfamiliar to the ICLR community.

Response. We have significantly revised Section 3, specifically
•	Clearly distinguished the discriminator (which checks plausibility under the world model) from the classifier (that detects out-of-distribution trajectories via the KS statistic).
•	Added Figure 1 summarizing the full pipeline.
•	Added a simple two-variable TC example in Appendix A.

2. Theoretical Guarantees / Identifiability

Reviewer concern. Missing formal justification for causal identifiability.

Response. We added a proposition showing that adding or removing key physics terms necessarily changes the system’s TC, establishing local causal identifiability. The proposition connects TC preservation directly to the causal structure of the governing equation.

3. Scope of Experiments & Benchmark Methods

Reviewer concern. The paper only considered two systems and lacks comparison with state-of-the-art methods, including causal learning methods.

Response. The original submission already contained five dynamical systems plus a real-world ball-drop example, including noisy cases. To demonstrate generalizability, we
•	Added experiments on 10 additional 3D/4D systems from ODEBench dataset.
•	Extended comparisons to Weak SINDy, ODEFormer, and SPL.
•	Presented a new metric called Dice Equation Similarity (DES) to demonstrate how CanDiCE is able to accurately identify the “right” physics terms as opposed to just fitting the noisy data. On average, the DES score for CANDiCE is 0.67 as compared to 0.47 for Ensemble SINDy, 0.50 for Weak SINDy, 0.45 for ODEFormer, and 0.55 for SPL.
•	We explained how traditional causal inference methods are not applicable because dynamical systems are generically non-separable, which motivated the development of Topological Causality.

4. Ablation / Sensitivity Analysis

Reviewer concern. How sensitive is the method to the sparsity and TC regularization terms?

Response. An ablation study was added in Appendix I showing how DES, RMSE, and TC behave when terms are progressively removed. The analysis reveals a sharp change in TC when a truly causal term is removed, while RMSE alone does not distinguish between sparsity patterns. This motivates using TC to separate causal from merely predictive terms.

5. Computational Cost and Scalability

Reviewer concern.  The method may not scale due to GAN training and TC computation.

Response.
•	Table 1 now includes runtime comparisons with Weak SINDy and ODEFormer.
•	Slowest experiment finishes in ~35 minutes on CPU.
•	TC evaluation is fast (<0.4 seconds even for the largest ODEBench systems).
•	Main cost is world-model construction via MCMC, not GAN sampling.
•	Regarding scalability to very high dimensions, we showed that the main scalability bottleneck lies in constructing the initial world model—not in GAN refinement—with Lorenz-96 experiments demonstrating strong performance up to 20 dimensions but degradation at 40 due to library densification, and clarified that our contribution is the use of topological causality to suppress non-causal terms rather than solving the broader open challenge of scalable world-model discovery.

---

### Meta-Review · Area_Chair_Bp1a · 2026-01-05

**Summary:**

The paper introduces a method for data-driven equation discovery in dynamical systems that explicitly exploits causal relationships between system variables. The authors start from the observation that existing approaches based on symbolic regression or sparse regression primarily rely on correlations rather than causality, and that classical causal inference methods are not well suited to dynamical systems. To address this limitation, they propose a novel methodology building on recent developments in topological causality and integrating counterfactual reasoning.
The approach consists of three main steps. First, a world model is learned in the form of a joint probability distribution over the coefficients of a predefined library of basis functions. Second, the method generates counterfactual explanations, i.e., alternative sets of coefficients that produce out-of-distribution trajectories relative to the observed data. These counterfactuals are used to identify causal relationships between variables. Finally, a constrained sampling mechanism is introduced to generate counterfactual instances while enforcing causal consistency and other structural constraints. The proposed framework is evaluated on several benchmark datasets and compared against representative methods from the symbolic regression and sparse regression literature.

The reviewers acknowledge the relevance of the problem and the originality of the proposed approach, particularly the formulation of causal consistency and its implementation via counterfactual generation. They nevertheless raise several concerns, including the computational complexity of the method and its ability to scale to larger problems, missing baselines and benchmarks, and the dependence of the identification process on the predefined basis library. Opinions on the paper’s organization and technical presentation are mixed: some reviewers find it adequate, while others consider it overly complex and difficult to follow.

**Reviewer Concerns:**

While the additional explanations improve clarity, the technical presentation remains overly complex and not pitched at an appropriate level for the intended audience. The paper would require a substantial reorganization to simplify the main text, remove unnecessary details, and clearly separate core ideas from specialized discussions, with detailed material deferred to the appendices. In its current form, the manuscript is difficult to follow and is likely to be accessible only to a very narrow audience.

**Reviewer Scores:**

RCSxc (rating 6) appreciates the contribution and is unlikely to change their score.

RxPLj (rating 2) is unlikely to change their assessment.

RhwVS (rating 4) may increase their score.

RH6Lp (rating 4) remains extremely concerned about the style and clarity of the technical description and is unlikely to change their rating.

---

### Decision · Program_Chairs · 2026-01-26

Reject